# Intracellular lipid droplet accumulation occurs early following viral infection and is required for an efficient interferon response

E. A. Monson [1], K. M. Crosse[1], M. Duan[2], W. Chen [2], R. D. O'Shea [1], L. M. Wakim[3], J. M. Carr [4], D. R. Whelan[2] & K. J. Helbig [1✉]

Lipid droplets (LDs) are increasingly recognized as critical organelles in signalling events, transient protein sequestration and inter-organelle interactions. However, the role LDs play in antiviral innate immune pathways remains unknown. Here we demonstrate that induction of LDs occurs as early as 2 h post-viral infection, is transient and returns to basal levels by 72 h. This phenomenon occurs following viral infections, both in vitro and in vivo. Virally driven in vitro LD induction is type-I interferon (IFN) independent, and dependent on Epidermal Growth Factor Receptor (EGFR) engagement, offering an alternate mechanism of LD induction in comparison to our traditional understanding of their biogenesis. Additionally, LD induction corresponds with enhanced cellular type-I and -III IFN production in infected cells, with enhanced LD accumulation decreasing viral replication of both Herpes Simplex virus 1 (HSV-1) and Zika virus (ZIKV). Here, we demonstrate, that LDs play vital roles in facilitating the magnitude of the early antiviral immune response specifically through the enhanced modulation of IFN following viral infection, and control of viral replication. By identifying LDs as a critical signalling organelle, this data represents a paradigm shift in our understanding of the molecular mechanisms which coordinate an effective antiviral response.

[1] Department of Physiology, Anatomy and Microbiology, School of Life Sciences, La Trobe University, Melbourne, VIC, Australia. [2] La Trobe Institute for Molecular Science, La Trobe University, Melbourne, VIC, Australia. [3] Department of Microbiology and Immunology, University of Melbourne, at Peter Doherty Institute for Infection and Immunity, Melbourne, VIC, Australia. [4] Microbiology and Infectious Diseases, College of Medicine and Public Health, Flinders University, Adelaide, SA, Australia. ✉email: k.helbig@latrobe.edu.au

L ipid droplets (LDs) are storage organelles that can modulate lipid and energy homoeostasis, and historically, this was considered their defining role. More recently, LDs have emerged as a dynamic organelle that frequently interacts with other organelles and are involved in protein sequestration and transfer between organelles. LDs have also been demonstrated to act as a scaffolding platform to regulate signalling cascades, highlighting their diverse functions[1–4].

The role of LDs in an infection setting has not been well studied, however, it has been demonstrated that LDs accumulate in leucocytes during inflammatory processes, and they are also induced in human macrophages during bacterial infections[2]. Multiple bacterial strains, including *Mycobacterium* spp., *Chlamydia* spp., *Klebsiella* spp. and *Staphylococcus* spp. are known to upregulate LDs very early following bacterial infection in both primary and cell culture macrophage models, and this has also been seen for a number of bacterial species in rodent macrophage cell lines[5–7]. Interestingly, *Trypanosoma cruzi* infection of macrophages also induces LDs, however, this response takes 6–12 days to occur following infection[8]. Bacterial- induced LD induction in immune cells has been shown to depend on toll-like receptor engagement, mainly via TLR2 and TLR4, however, the role of LDs in the outcome of bacterial infection remains largely unknown, and the exact mechanisms for controlling LD induction remain elusive[9,10]. It has been suggested in recent work in the zebrafish model that embryos with higher levels of LDs are more protected from bacterial infections[11] and work in the Drosophila embryo has demonstrated that LDs can bind to histones which are released upon detection of intracellular bacterial LPS and act in a bactericidal manner[12].

Interestingly, LD induction has been demonstrated to be a direct result of immune activation of macrophages by IFN-γ in a HIF-1α dependent signalling pathway[13]. *M. tuberculosis* acquires host lipids in the absence of LDs under normal conditions, however, IFN-γ stimulation of macrophages results in redistribution of host lipids into LDs where *M. tuberculosis* is unable to acquire them[13]. IFN-γ induced LDs have also been shown to enhance the expression of genes involved in LD formation and clustering in INS-1β cells. More importantly, pre-treatment of INS-1β cells with IFN-γ markedly increased PIC-induced expression of antiviral genes (e.g., *Ifnb, Mx1*)[14].

Although induction of LDs has been documented to occur mainly in macrophage models, following infection with bacteria, the ability of viral infection of cells to induce the same response remains relatively unexplored. Recently, viral infection of the positive-stranded RNA viruses, Sindbis and dengue virus, was shown to induce LD formation in the cells of mosquito midgut[15]. This LD induction was mimicked via synthetic activation of the antiviral innate pathways, Toll-Like and immune deficiency (IMD), similar to the induction of bacterial-induced LDs. Although it is known that activation of early innate signalling pathways appears to induce LDs in the presence of bacteria, and in the mosquito midgut when virally infected, the mechanisms at play remain unknown, as does the functional outcome of this LD induction.

Here we show that induction of LDs is a transient phenomenon following viral infection both in vitro and in vivo, and occurs 2 h following virus recognition, with a return to baseline at 72 h. Virally induced LD induction is type-I IFN independent, but dependent on EGFR engagement and corresponds with enhanced cellular production of both type-I and -III IFN in infected cells. In this work enhancing LD numbers prior to viral infection positively modulates the host cell antiviral state and decreases HSV-1 and ZIKV replication. Collectively, our results suggest that LDs play a pivotal role in facilitating the magnitude of the early innate antiviral immune response.

## Results

**Lipid droplets are induced early following viral infection**. To determine if LD induction following viral infection is a common phenomenon in mammalian cells, we infected cultured cells with viruses from 3 different viral families. Herpes simplex virus-1 (HSV-1), influenza A (IAV), dengue virus (DENV) and zika virus (ZIKV) all induced upregulation of LDs at 8 h following infection, as seen via microscopy (Fig. 1 and Supplementary Fig. 1). IAV infection of THP-1 monocytes with either the virulent PR8 strain or the more attenuated X-31 strain induced a 6.5-fold increase in LD numbers (Fig. 1a, c). Primary human foetal immortalised astrocytes were assessed for their ability to upregulate LDs when infected with the neurotropic viruses ZIKV, DENV and HSV-1. Astrocytes were seen to have a high average basal level of LDs per cell (~15 per cell) (Fig. 1b, c), which was significantly increased by 3.9-, 5.3- and 4-fold following infection of these cells with either ZIKV, DENV or HSV-1, respectively (Fig. 1b, c and Supplementary Fig. 1A). In vivo, we examined lung sections taken from both mock and IAV infected C57BL/6 mice. A clear increase in the presence of large LDs was detected near the bronchioles in the IAV infected mice as well as an increase in small LDs throughout the tissue, which was absent in the mock-infected mice at both 1- and 3-days post-infection (Fig. 1d and Supplementary Fig. 1B). The size variation seen in LDs in these tissues may reflect the different cell types recruited to the site of infection. We also examined DENV infection in vivo in 1-day old BALB/c pups, with an increase of LDs visualised in the brain and back of the eye where DENV viral RNA was detected with this accumulation absent in mock-infected pups (Fig. 1e and Supplementary Fig. 1B).

HSV-1, ZIKV, DENV and IAV viruses enter their host cell by either plasma membrane fusion or following endocytosis, prior to the release of their genomic material[16,17]. In order to determine if pattern recognition receptor (PRR) detection of nucleic acid alone would drive an induction of LDs in cells, we stimulated these cells with the synthetic viral mimics, dsRNA (poly I:C, known to mimic viral RNA pathogen associated molecular patterns (PAMPs), and activate the RNA sensors RIG-I and TLR3) or dsDNA (poly dA:dT, known to mimic DNA viral PAMPs, and activate cytosolic DNA sensors). In addition, the use of these viral mimics may allow the further dissection of the mechanisms of LD induction following activation of viral PRRs in the absence of viral antagonism. As can be seen using confocal microscopy in Fig. 2a, rhodamine labelled dsRNA and dsDNA clearly induced an upregulation of LDs in primary immortalised astrocytes. To determine if this was a common phenomenon across cell types, similar experiments were performed in primary murine foetal astrocytes, THP-1 monocytes/macrophages, HeLa cells and primary murine embryonic fibroblasts (MEFs). Astrocytes were seen to have a high basal level of LDs, with primary foetal murine astrocytes and immortalized human astrocytes having an average of 22 and 18 LDs per cell respectively; this contrasted with the lower levels of LDs seen in other cell types, which ranged from 6 to 9 LDs per cell (Fig. 2b). All cell types stimulated with either of the viral mimics upregulated LDs at 8 h (Fig. 2b and Supplementary Fig. 2). Stimulation of cells with dsRNA resulted in LD upregulation fold changes ranging from 4.1-fold in the MEFs to 9.5-fold in the THP-1 macrophages (Fig. 2b). Similarly, dsDNA stimulation resulted in a 4.1-fold induction in HeLa cells and, up to a 10.2-fold induction in the THP-1 monocyte cells (Fig. 2a). This increase was also shown to be independent of whether FCS was in the culture media (Supplementary Fig. 2).

Although LD numbers increased in all cell types following stimulation with the viral mimics, the average size of LDs did not (Fig. 2c). The average basal size of LDs was consistent across most cell types, with a diameter range of 280–400 nm (Fig. 2c);

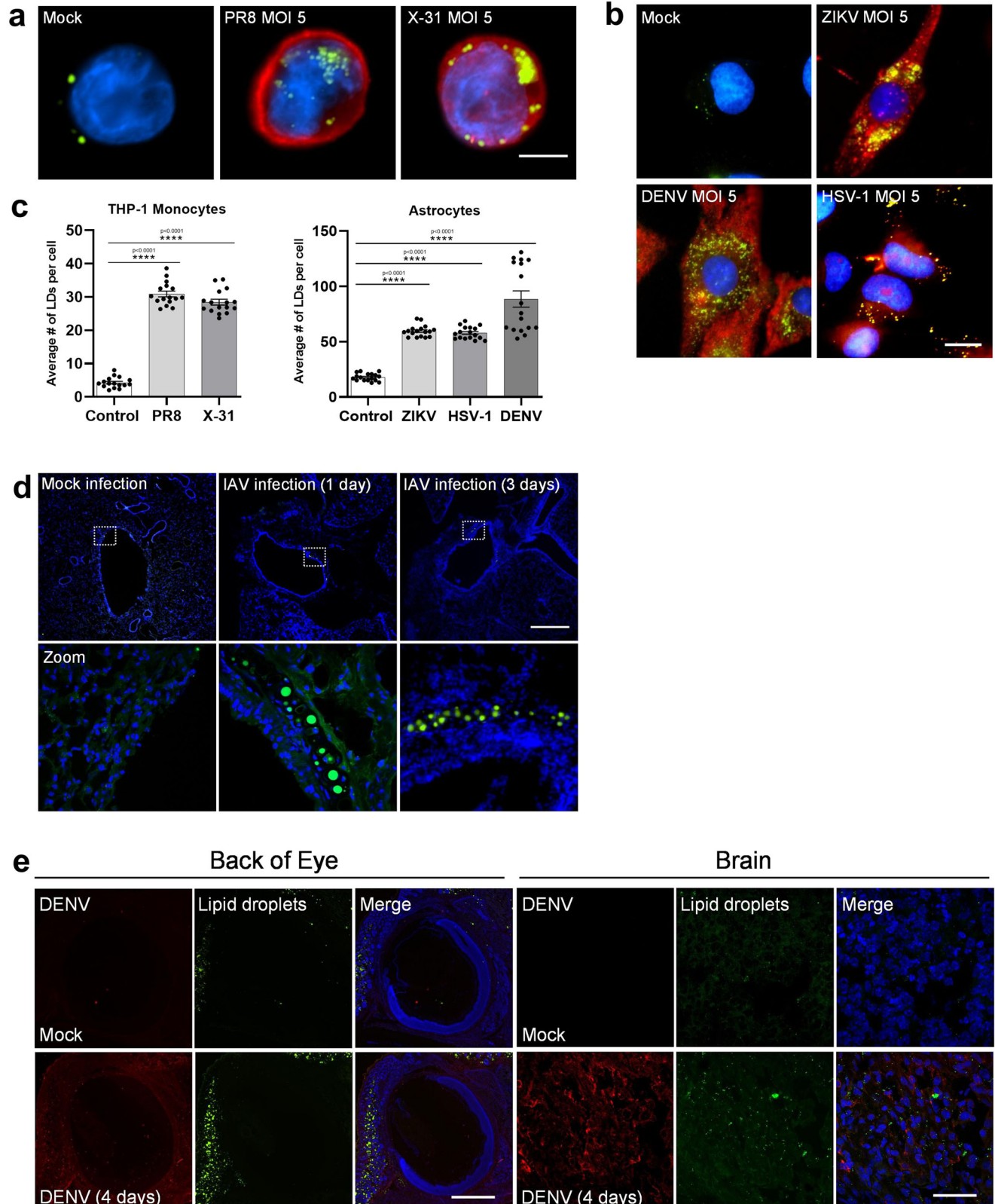

however, THP-1 macrophages had a starting average basal LD size of 3100 nm, which did not increase following stimulation with either dsRNA or dsDNA. In contrast, all other cell types had an increased average LD size at 8 h following dsRNA stimulation, ranging from a two-fold increase in THP-1 monocytes to a 5.3-fold increase in HeLa cells, with similar size increases observed

following dsDNA stimulation also (Fig. 2c). The average size of LDs in the primary immortalised astrocytes following stimulation with viral mimics ranged from 760 to 910 nm (Fig. 2c), however as can be seen in Fig. 2d, there was a significant increase in the number of LDs greater than 1000 nm in these cells, and also, a substantial increase in LDs less than 200 nm, which are referred

**Fig. 1 Lipid Droplets accumulate in response to IAV, ZIKV and HSV-1 infections. a** Human THP-1 monocytes were infected with two different strains of influenza- PR8 and X-31 at an MOI 5 for 8 h, Scale bars, 50 μm. **b** Primary immortalised astrocyte cells were infected with either the ZIKV (MR766 strain), DENV (DENV2) or HSV-1 (KOS strain) at MOI 5 for 8 h. All cells were stained with Bodipy (493/503) to visualise LDs (green) and DAPI to visualise the cell nuclei (blue). IAV was detected with a αNS2 antibody (1:1000), ZIKV and DENV RNA was detected using an anti-3G1.1 and 2G4 dsRNA antibodies (in combination, used neat) and HSV-1 was detected using the anti-HSV-1 antibody (Abcam, ab9533), all shown in red staining. Scale bars, 50 μm. **c** LD numbers were analysed using ImageJ analysis software. Error bars, mean values ± SEM, P values were determined by unpaired two-tailed Student's t test with a Holm-Sidak correction for multiple comparisons ($n = 2$ biological replicates). Stimulated cells were statistically compared with their respective mock controls. **d** C57BL/6 mice were either mock infected or infected with $10^4$ PFU of IAV for 24 or 72 h prior to removal of both lung lobes for immunofluorescence analysis of LDs via Bodipy (493/503) staining (green). DAPI was utilised to visualise the cell nuclei (blue), scale bars, 500 μm. **e** 1-day old BALB/c pups were either mock infected or infected with 800 PFU of DENV-2 (MON601) for 2 or 4 days prior to removal of pup heads for immunofluorescence analysis of LDs via Bodipy (493/503) staining (green) in the brain and eye. DAPI was utilised to visualise the cell nuclei (blue) and DENV RNA was detected using anti-3G1.1 and 2G4 dsRNA antibodies (in combination). Scale bars, 500 μm, $n = 3$ mice. Source data are provided as a Source Data file.

to as nascent LDs[18]. Nascent LDs made up 24 and 23% of the LD population following dsRNA and dsDNA stimulation respectively, in comparison to only 13% in control-treated cells. In addition, inhibition of triglyceride synthesis completely abrogated the upregulation of LDs following stimulation with dsRNA and dsDNA viral mimics (Supplementary Fig. 3A, B). Collectively, these results indicate that nucleic acid stimulation drives both the generation of new LDs as well as the growth of existing LD populations.

**Lipid Droplet accumulation is transient following detection of intracellular nucleic acids and follows a similar time course to interferon mRNA upregulation.** To define the dynamics of LD induction following the detection of nucleic acids in the cells, we set up a time course series to quantify the speed and longevity of this response. Astrocyte cells were chosen to further examine the role of LDs in viral infection due to their extensive upregulation of LDs following viral stimulation and infection. In addition, astrocytes are known to be rapid producers of an effective anti-viral response[19]. LDs were upregulated as early as 2 h following either dsRNA or dsDNA stimulation (Fig. 3a, b), stayed significantly upregulated for 48 h post-stimulation and returned to baseline levels by 72 h. The average LD number per cell increased from ~17 to 28–40 LDs per cell at 2 h post-stimulation, depending on the stimulation type. Interestingly, dsRNA or dsDNA stimulated cells reached a maximum LD induction between 4 and 8 h, however, dsRNA stimulated cells showed an initial decrease in LD number at 24 h and, a subsequent increase at 48 h, prior to returning to baseline levels at 72 h, indicating a biphasic response, which was not seen following stimulation of the cells with dsDNA. Average LD size per cell was also shown to transiently increase over the same time course (Supplementary Fig. 3C). Interestingly, the induction of LDs followed a similar pattern to the production of type-I and –III IFN mRNAs in the astrocyte cells (Fig. 3c, d), where peak IFN mRNA induction was seen at 8 h post dsDNA stimulation, but at 24 h after dsRNA stimulation. IFN mRNA levels showed a trend of returning to basal levels after 72 h.

**Increasing cellular LD numbers acts to enhance the type I and III IFN response to viral infection.** We have previously demonstrated that loss of cellular LDs impacts the host cell response to viral infection in vitro[20]. To determine if the upregulation of LDs following viral infection plays an anti-viral role in the cell, we initially established a LD induction model in the primary immortalised astrocytes. Addition of oleic acid to cells has previously been shown to enhance LDs minutes following treatment in Huh-7 cells, with LDs remaining upregulated for 3–4 days following addition of oleic acid to the media[21,22]. As can

be seen in Fig. 4a, b, the addition of 500 μM of oleic acid to astrocytes in cell culture for 16 h increased the average LD number from ~16 to 43 per cell. Furthermore, despite the increase in cellular LD numbers, stimulation of cells with either dsRNA or dsDNA was able to further upregulate cellular LD levels (Fig. 4c and Supplementary Fig. 4A). Interestingly, LD upregulation was accompanied by significantly enhanced IFN transcription and translation (Fig. 4d–g). In the presence of oleic acid enhanced LD numbers, a significant increase in IFN mRNA transcription was seen (Fig. 4d, e), although, no increase at the protein level (Fig. 4f, g). Addition of dsRNA to the cells in the presence of enhanced LDs (oleic acid treated) showed a twofold increase in IFN-β and IFN-λ mRNA at 8 h, which was accompanied by a two-fold increase in the mRNA of the interferon stimulated gene, viperin. However, increases in the transcriptional level for these genes were only observed at 24 h for IFN-λ and viperin (2 and 2.6-fold respectively; Fig. 4d, e). There was no increase the IFN-β transcriptional response in cells with an enhanced LD content which were treated with dsDNA, however, a small but significant increase in IFN-λ and viperin mRNA was observed at both 8 and 24 h post-stimulation (1.5 and two-fold respectively at 8 h, and 2.5 and two-fold at 24 h (Fig. 4d, e)). In confirmation of the transcriptional upregulation of IFNs, significantly enhanced protein levels could be seen for both IFN-β and IFN-λ following either dsRNA or dsDNA stimulation of primary immortalised astrocytes with oleic acid induced LDs, in comparison to controls (Fig. 4f, g). The presence of upregulated LDs was able to significantly enhance the production of IFN-β and IFN-λ protein by as much as 2.6 and 3.6-fold in the presence of dsRNA and 2.0 and 2.1-fold in the presence of dsDNA. Interestingly, the production of both IFN-β and IFN-λ was much greater following stimulation with dsDNA in comparison to dsRNA in the astrocyte, with IFN-λ being the dominantly expressed IFN species.

Next, we assessed the host antiviral response to viral infection, in the presence of enhanced LDs. First, we confirmed the ability of astrocyte cells to induce LDs further once lipid loaded from oleic acid treatment (Fig. 5a and Supplementary Fig. 4B). LD loaded cells, when challenged with ZIKV demonstrated a 3.5-fold increase in the production of IFN-β mRNA at 24 h and a small but significant increase of 1.7-fold at 48 h post-infection when compared with control infected cells (Fig. 5b). IFN-λ followed a similar trend showing a 3.3 and a 2.2-fold increase at 24 and 48 h, respectively (Fig. 5b), and a 5-fold increase in IFN-λ mRNA at just 6 h post-infection. Interestingly, when looking at the production of a key antiviral signalling and LD resident protein, viperin, cells with enhanced LDs showed a significant increase in mRNA at 6, 24- and 48-h post ZIKV infection. Cells infected with the dsDNA virus, HSV-1 also showed a similar trend, where the production of mRNA for both IFN-β and IFN-λ as well as viperin were enhanced in cells pre-treated with oleic acid (Fig. 5c). These

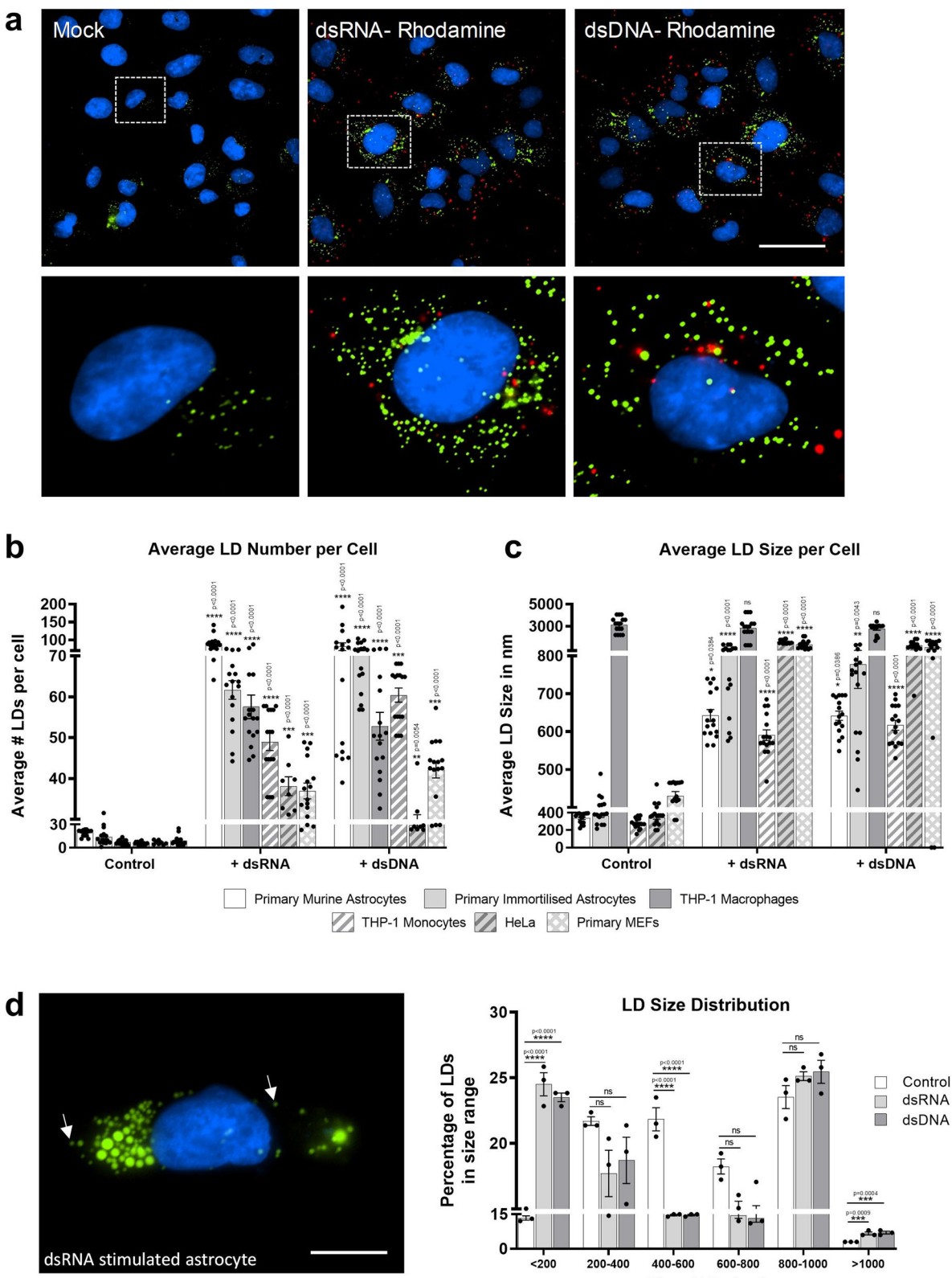

results correlated well with a reduced viral load of both ZIKV and HSV-1 at 24 h (6.3-fold and 2.3-fold for ZIKV and HSV-1 respectively) (Fig. 5d, e) and at 48 h post infection (1.4-fold decrease in ZIKV mRNA and a 2.6-fold decrease in HSV-1 (Fig. 5d, e)). This reduction in viral load for ZIKV coincided with a significantly enhanced level of both IFN-β and

IFN-λ production by the astrocytes with upregulated LDs (Fig. 5f).

**Lipid droplets accumulate in response to IFN, despite initial accumulation being type-I IFN independent.** Detection of aberrant nucleic acid in cells drives a rapid interferon response[23].

**Fig. 2 Detection of intracellular dsRNA and dsDNA initiates accumulation of LDs in multiple cell types. a** Primary immortalised human astrocyte cells stimulated with dsRNA and dsDNA tagged with Rhodamine (red) for 8 h and stained with Bodipy (493/503) to visualise LDs (green) and DAPI to visualise the cell nuclei (blue). Cells were imaged on a Nikon TiE microscope. Original magnification is 60X. Scale bar, 50 μm. **b** Average number of LDs per cell and (**c**) average LD sizes (diameters) were analysed from greater than 200 cells in a range of cell types, using ImageJ analysis software ($n = 2$ biological replicates). **d** LD size distribution in primary immortalised astrocyte cells stimulated with either dsDNA or dsRNA for 8 h. Image represents a single astrocyte cell following dsRNA stimulation. Cells were stained with Bodipy (493/503) to visualise LDs (green) and DAPI to visualise the cell nuclei (blue) and LD size distributions were analysed on ImageJ analysis software. Scale bar, 15 μm. In (**b**–**d**) error bars, mean values ± SEM, $P$ values were determined by unpaired two-tailed Student's $t$ test with a Holm-Sidak correction for multiple comparisons (greater than 300 cells; $n = 3$ biological replicates). Stimulated cells were statistically compared with their respective mock controls, ns = not significant. Source data are provided as a Source Data file.

In order to determine if LD induction following the detection of intracellular nucleic acids required the production of IFN, we stimulated Vero cells, which lack the ability to produce IFN due to spontaneous gene deletions[24,25], with both dsRNA and dsDNA. Both LD number and size were significantly upregulated in Vero cells at 8 h post-stimulation (Fig. 6a, b and Supplementary Fig. 5), indicating that this is an IFN independent event. As can be seen in Fig. 6c, d, LDs were significantly induced by up to 4.5-fold following interferon stimulation. To show this in a more physiologically relevant setting, astrocyte cells were treated with dsRNA and dsDNA and left to produce IFNs for 24 h, and their conditioned media was removed and placed on untreated astrocyte cells. Conditioned media from cells stimulated with dsRNA was also shown to induce LDs by 6.3-fold, a similar level to that induced by 1000 U/mL of IFN-β (Fig. 6e). Interestingly, conditioned media from cells stimulated with dsDNA showed no increase in LD numbers (Fig. 6e), perhaps indicating the presence of an inhibitor of LD induction. To confirm that it was the presence of secreted IFNs in the conditioned media alone, that was driving the production of LDs, we took conditioned media from both dsRNA and dsDNA stimulated astrocytes and Vero cells at 24 h following stimulation and placed it back onto untreated cells. As the Vero cells lack the ability to secrete type-I IFNs, we expected to see no induction of LDs in cells receiving conditioned culture media from these cells, which we observed (Fig. 6e). The induction of LDs was only driven with the addition of dsRNA conditioned media removed from astrocytes and placed onto both naive astrocytes and Vero cells. Interestingly, the addition of conditioned culture media from Vero cells stimulated with dsDNA onto untreated astrocytes cells showed a 2.7-fold decrease in the average number of LDs per cell relative to control untreated cells (Fig. 6e). Perhaps, further demonstrating the presence of a secreted negative regulator of LD biogenesis following dsDNA stimulation of astrocytes.

To fully elucidate the role IFNs were playing in LD accumulation, we utilised an antibody that could block IFN inducible LDs in the astrocytes (Supplementary Fig. 5B). To validate the bi-phasic and IFN independent LD accumulation following dsRNA stimulation, we blocked IFNAR1 in astrocyte cells and stimulated them with dsRNA and dsDNA for up to 72 h (Fig. 6f and Supplementary Fig. 5C). There was no difference in LD accumulation at earlier time points in cells that were treated with antibody controls compared with IFNAR1 blocking antibodies, however, when looking at the 48 h time point following stimulation, the IFNAR antibody partially inhibited the second wave of LD induction seen in the dsRNA stimulated cells, with a small but significant drop in the number of LDs also in the dsDNA stimulation condition (Fig. 6f and Supplementary Fig. 5C). These results further support initial IFN independent induction of LDs following both dsRNA and dsDNA stimulation, with a second upregulation of LDs being dependent on IFN signalling.

**LD induction following nucleic acid detection is EGFR mediated.** Phospholipase A2 (PLA₂) is an enzyme known to be a key player in LD biogenesis, where it catalyses the hydrolysis of glycerophospholipids to release fatty acids from phospholipid membranes which are then sequestered into the ER membrane leading to the maturation and budding off of mature LDs[18]. Astrocyte cells were treated with AACOCF₃, a well-described inhibitor of PLA₂[26] and their ability to induce LDs was assessed. AACOCF₃ was able to inhibit LD biogenesis post serum starvation (Supplementary Fig. 6A, B), confirming that natural LD biogenesis in astrocytes requires PLA₂ activation. To assess whether LD induction following recognition of viral mimics also follows a PLA₂ driven mechanism, cells were treated with AACOCF₃ prior to stimulation. Inhibition of PLA₂ did not inhibit the induction of virally induced LDs in the primary immortalized astrocyte cells (Fig. 7a, b). EGFR engagement has previously been shown to control LD upregulation in colon cancer[27]. To assess whether EGFR was important in LD biogenesis following viral mimic stimulation, primary immortalized astrocyte cells were treated with AG-1478, a well-described tyrosine kinase inhibitor of EGFR[28] and stimulated with dsRNA and dsDNA to evaluate LD induction. Astrocyte cells treated with AG-1478 demonstrated no induction of LDs after stimulation with dsRNA or dsDNA, however, AG-1478 did not inhibit the induction of LDs following oleic acid treatment, with LDs being induced approximately fivefold (Fig. 7c, d). Similarly, the treatment of MCF-7 cells (known to lack EGFR[29]), also resulted in no upregulation of LDs following stimulation with viral mimics but was able to upregulate LDs in the presence of oleic acid (Supplementary Fig. 7A, B). However, the inhibition of EGFR did not alter LD biogenesis post serum starvation (Fig. 7e), indicating that the EGFR receptor is able to mediate the induction of viral mimic driven LDs, but not natural biogenesis of LDs in astrocytes. Further downstream analysis also demonstrated that the EGFR mediated induction of virally driven LDs relies on subsequent PI3K activation in the cell (Supplementary Fig. 7).

A time course of LD induction in cells treated with AG-1478 demonstrated that at 8 h, there is no LD induction, confirming that the initial upregulation of LDs following nucleic acid stimulation is dependent on EGFR. However, at 24 h post-stimulation, there was a 2.5-fold increase in LD numbers in dsRNA stimulated cells, but not in dsDNA stimulated cells (Fig. 7f). At 48 h post-stimulation, a similar trend was observed with a fourfold induction in the dsRNA stimulated cells, but again no LD induction in the dsDNA stimulated cells (Fig. 7f). This result may explain the biphasic expression pattern of LDs seen following dsRNA stimulation of astrocytes, but not dsDNA stimulation (Fig. 3b), particularly if the second wave of LD induction is not dependent on EGFR. To assess this, we treated primary immortalised astrocyte cells with AG-1478 to inhibit EGFR and stimulated them with IFN-β and analysed their LD numbers after 16 h. There was no significant difference in the upregulation of LDs of control cells compared with cells treated with AG-1478 when stimulated with IFN-β indicating EGFR does not play a role in the upregulation of LDs induced with IFN stimulation (Fig. 7g).

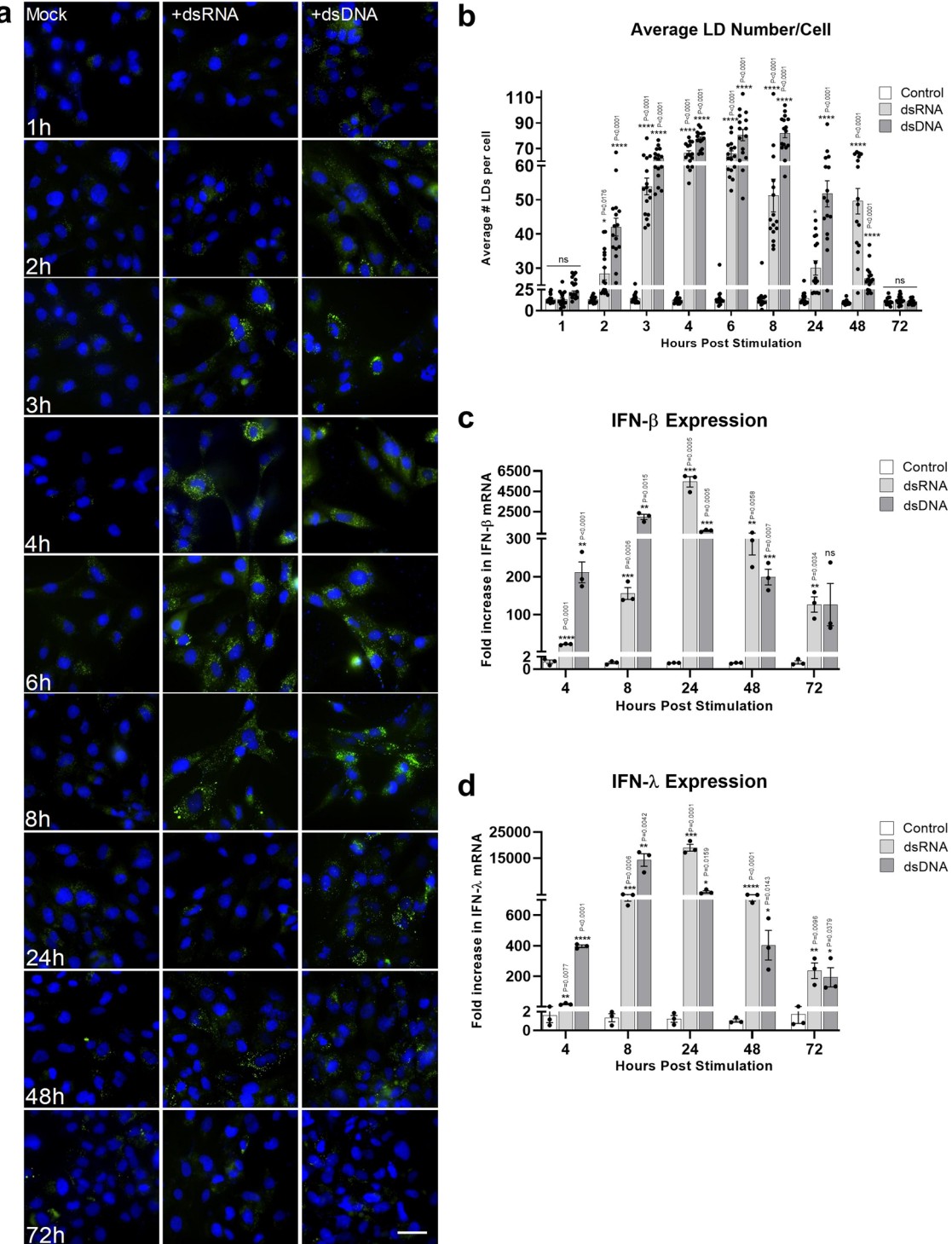

**Fig. 3 Lipid Droplet accumulation is transient following detection of intracellular nucleic acids. a** Primary immortalised astrocyte cells were stimulated with dsRNA and dsDNA and were fixed at regular time points until 72 h post-stimulation. Cells were stained with Bodipy (493/503) to visualise LDs (green) and DAPI to visualise the cell nuclei (blue). Scale bar, 50 μm. Images are a representation of $n = 3$ independent experiments. **b** Average number of LDs per cell were analysed from all time points using ImageJ analysis software ($n > 300$ cells; $n = 3$ separate biological replicates). **c**, **d** Primary immortalised astrocyte cells were stimulated with dsRNA and dsDNA and RTq-PCR was utilised to quantify IFN-β and IFN- λ mRNA up to 72 h post-stimulation. In (**b**–**d**) error bars, mean values ± SEM, P values were determined by unpaired two-tailed Student's t test with a Holm-Sidak correction for multiple comparisons (greater than 300 cells; $n = 3$ biological replicates). Stimulated cells were statistically compared with their respective mock controls, ns = not significant. Source data are provided as a Source Data file.

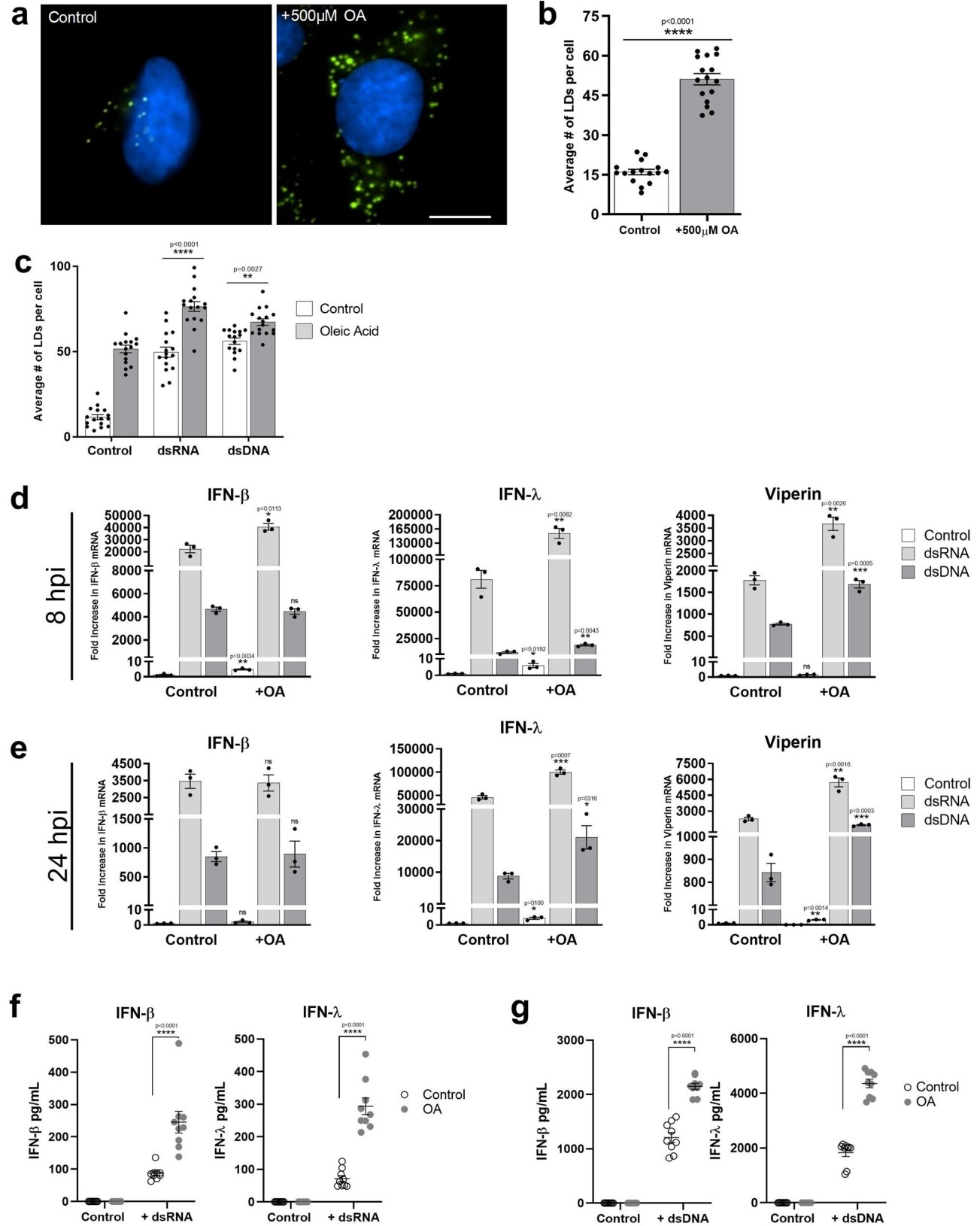

## Inhibition of EGFR driven LDs impacts IFN production and attenuates viral infection

EGFR is a receptor tyrosine kinase that is commonly upregulated in cancers but is also targeted by several viruses[30]. As it is known that EGFR underpins LD upregulation in some cancers, and in Fig. 7 we have demonstrated that EGFR could also drive virally induced LDs. We next wanted to understand the relationship between viral-induced EGFR driven LD biogenesis and the regulation of antiviral cytokines.

Primary immortalised astrocytes were pre-treated with AG-1478 prior to being stimulated with dsRNA and dsDNA, and their ability to upregulate IFN mRNA assessed. Both IFN-β, IFN-λ and viperin mRNA levels were significantly downregulated at 8 h post

**Fig. 4 Increasing cellular LD numbers acts to enhance the type I and III IFN response to dsRNA and dsDNA. a, b** Primary immortalised astrocyte cells were treated with 500 μM oleic acid (OA) for 16 h. Cells were stained with Bodipy (493/503) to visualise LDs (green) and DAPI to visualise the cell nuclei (blue) and LDs numbers were assessed with ImageJ analysis software (greater than 200 cells, $n = 2$) Bar, 15 μm. **c** Primary immortalised astrocyte cells were treated with 500 μM oleic acid for 16 h prior to stimulation with dsDNA or dsRNA for 8 h LD numbers were analysed. RT-qPCR was performed to evaluate IFN-β, IFN-λ and viperin mRNA expression at (**d**) 8 h or (**e**) 24 h post stimulation. All results are in comparison to RPLPO expression. In (**b–e**) error bars, mean values ± SEM, P values were determined by unpaired two-tailed Student's t test with a Holm-Sidak correction for multiple comparisons for 2 or more groups (greater than 300 cells; $n = 3$ biological replicates). Stimulated cells were statistically compared with their respective mock controls, ns = not significant. **f, g** IFN protein levels in the media from the previous experiments at 16 h post-infection were analysed via ELISA for IFN-β and IFN-λ protein. Error bars, mean values ± SEM, P values were determined by two-way ANOVA post-hoc pairwise comparisons with Bonferroni correction ($n = 3$ biological replicates). Source data are provided as a Source Data file.

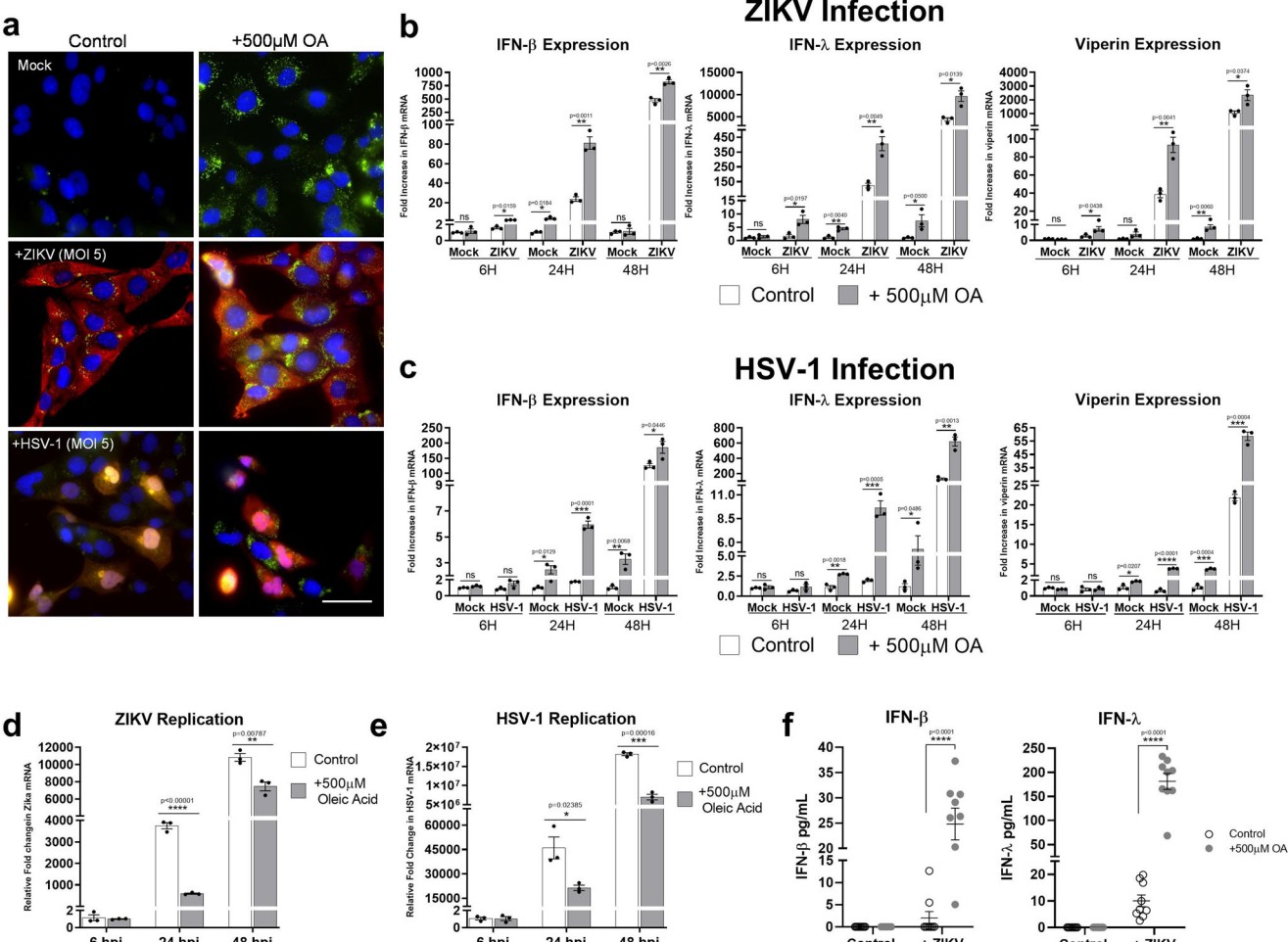

**Fig. 5 Increasing cellular LD numbers enhances IFN responses to restrict ZIKV and HSV-1 viral replication. a** Primary immortalised astrocyte cells were treated with 500 μM oleic acid (OA) for 16 h prior to infection with ZIKV (MR766 strain) at MOI 0.1 or HSV-1 (KOS strain) at MOI 0.01 for 8 h. Cells were stained with Bodipy (493/503) to visualise LDs (green) and DAPI to visualise the cell nuclei (blue). ZIKV RNA was detected using an anti-3G1.1 and 2G4 dsRNA antibodies and HSV-1 was detected using the anti-HSV-1 antibody (Abcam, ab9533), both viral proteins shown with red staining. Images are a representation of $n = 3$ independent experiments. RT-qPCR was utilised to evaluate IFN-β, IFN-λ and viperin mRNA expression at 8, 24 and 48 hpi post (**b**) ZIKV or (**c**) HSV-1 infection (MOI 0.1). Primary immortalised astrocyte cells were treated with 500 μM oleic acid for 16 h prior to infection with (**d**) ZIKV at a MOI 0.1 or (**e**) HSV-1 at an MOI 0.1, and RT-qPCR was utilised to evaluate viral replication at 6, 24 and 48 hpi. In (**b–e**) error bars, mean values ± SEM, P values were determined by unpaired two-tailed Student's t test with a Holm-Sidak correction for multiple comparisons for 2 or more groups (greater than 300 cells; $n = 3$ biological replicates). Stimulated cells were statistically compared with their respective mock controls, ns = not significant. **f** At 16 h post-infection secreted IFN protein levels from these experiments were analysed with ELISA plates for IFN-β and IFN-λ protein. Error bars, mean values ± SEM, P values were determined by two-way ANOVA post-hoc pairwise comparisons with Bonferroni correction ($n = 3$ biological replicates). Source data are provided as a Source Data file.

nucleic acid treatment, with little change being present at 24 h post-stimulation (Supplementary Fig. 8A). However, there was a significant difference in IFN-β protein detected in these samples with a dampened IFN response in cells with EGFR blocked

(Fig. 7h). The results were more pronounced when comparing IFN mRNA induction following both ZIKV and HSV-1 infection. Inhibition of EGFR inhibited the astrocytes ability to upregulate LDs when infected with either ZIKV or HSV-1 (Fig. 8a and

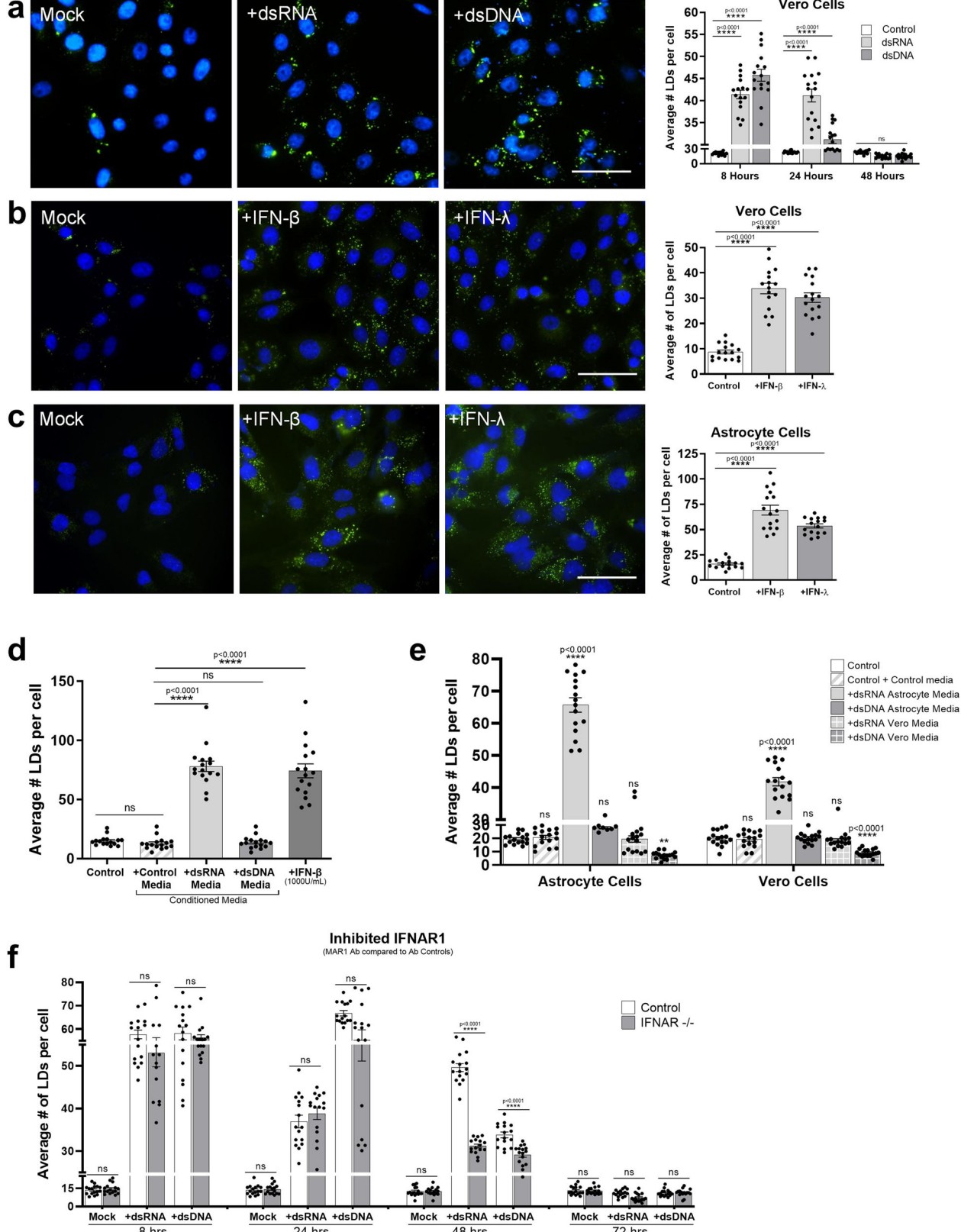

Supplementary Fig. 8B). This inhibition of EGFR driven LDs did not impact the ability of ZIKV or HSV-1 to enter astrocytes, as evidenced by the comparisons of PCR results for 6-hour time points for both viruses as well as immunofluorescent staining of ZIKV and HSV-1 proteins in astrocytes (Fig. 8a–c and Supplementary Fig. 8B); however viral replication was enhanced by as much as 26 and 24-fold at 24 h post infection, and 2 and 24-fold at 48 h post-infection with ZIKV and HSV-1 respectively. In addition, heightened viral nucleic acid levels corresponded to significantly lowered mRNA and protein levels of IFN-β at both the 24 and 48 h time points for ZIKV and HSV-1 infection (Fig. 8b, c and f) as well as significantly reduced IFN-λ mRNA

**Fig. 6 Lipid droplet accumulation following intracellular nucleic acid detection is type I IFN Independent. a** Vero cells were stimulated with dsRNA and dsDNA and were stained with Bodipy (493/503) to visualise LDs (green) and DAPI to visualise the cell nuclei (blue) at 8 h post-stimulation, bars, 50 μm. Cells were fixed at 8, 24 and 48 h post stimulation and analysed for LD numbers using ImageJ analysis software (greater than 200 cells (n = 2)) (**b**) Vero cells were stimulated with either IFN- β or IFN-λ for 8 h prior to fixation and staining with Bodipy (493/503) to visualise LDs (green) and DAPI to visualise the cell nuclei (blue), bars, 50 μm. LD numbers were analysed using ImageJ analysis software (greater than 200 cells (n = 2)). **c** Primary immortalised astrocyte cells were stimulated with either IFN- β or IFN-λ for 8 h prior to fixation and staining, and LD analysis, all performed as above (greater than 200 cells (n = 2)). **d** Astrocyte cells were treated with pre-conditioned media from prior dsRNA or dsDNA stimulated astrocyte cells or were stimulated with 1000 U/mL of INF-β for 8 h and their LD numbers were analysed using ImageJ analysis software (greater than 200 cells (n = 2)). **e** Astrocyte and Vero cells were treated with dsRNA or dsDNA conditioned media from either astrocytes or Vero cells and their LD numbers were analysed using ImageJ analysis software (greater than 200 cells (n = 2)). **f** Astrocyte cells were treated with MAR1 (anti IFNAR1) antibody to block type-I IFN signalling. Cells were then stimulated with dsRNA and dsDNA up to 72 h and LDs were analysed using ImageJ analysis software. In (**a**–**f**) error bars, mean values ± SEM, P values were determined by unpaired two-tailed Student's t test with a Holm-Sidak correction for multiple comparisons for 2 or more groups (greater than 300 cells; n = 3 biological replicates). Stimulated cells were statistically compared with their respective mock controls, ns = not significant. Source data are provided as a Source Data file.

levels for ZIKV at both time points, and at 24 h post-infection following HSV-1 infection. There was no IFN-λ expression observed at 48 h following HSV-1 infection. The production of both type I and III IFN mRNA levels also corresponded to the production of mRNA levels for the interferon stimulated gene viperin, with significantly lowered mRNA levels seen in cells treated with the EGFR inhibitor prior to viral infection. These results are indicative of a reduced ability of the cell to produce IFN following viral infection when LD induction is inhibited using the EGFR kinase inhibitor, AG-1478.

## Discussion

Lipid droplets are well known for their capacity as lipid storage organelles, however, more recently, they have emerged as critical organelles involved in numerous other biological functions. LD biology is an emerging field, with recent discoveries describing roles for LDs in multiple signalling and metabolic pathways as well as protein-protein and inter-organelle interactions[1,3,4]. LDs are now considered a highly dynamic organelle involved in facilitating multiple cellular pathways and responses, however, their role in immunity remains relatively unexplored. We have previously shown that loss of LD mass impairs the antiviral response and enhances viral replication[20], however, the dynamic induction of LDs and the mechanism responsible for this, as well as their role in the innate immune signalling response, has not previously been characterised.

It has previously been described that the accumulation of LDs can occur in leucocytes during inflammatory processes, and that LDs are induced by a number of bacterial infections in macrophages (reviewed in[2]). The mechanisms behind such induction have been shown to be dependent on toll-like receptor engagement, however, their role in the outcome of bacterial infection is not known, and the exact mechanisms required for their induction remains elusive[2]. Recently, a role for LDs in the antiviral response was proposed for the mosquito, when viral infection was shown to induce LD formation in the cells of the midgut[15]. As this is a phenomenon that has never been observed in mammalian biology, we sought to understand how and why LDs were induced following viral infection.

We analysed the dynamic induction of LDs post activation of innate signalling pathways in a number of cell types, both primary and non-primary, to assess their ability to induce LDs upon infection. LDs were induced upon infection with ZIKV, IAV or HSV-1 (Fig. 1a–c) in an in vitro setting, as well as early in vivo following influenza infection in a murine model (Fig. 1d). Interestingly, members of the *Flaviridae* family of viruses (HCV, ZIKV and DENV) have previously been demonstrated to deplete LDs by utilising fatty acids to facilitate aspects of their viral life cycle[31,32], with HCV and DENV also utilising LDs as a

platform for viral assembly, where they induce their lipolysis, and manipulate their biogenesis (reviewed in[33]). Recently, Laufman et al. (2019) also demonstrated a relationship for enteroviruses with LDs, where replication complexes were shown to tether to LDs via viral proteins, to subvert the host lipolysis machinery, enabling the transfer of fatty acids from LDs and leading to the depletion of LDs in infected cells[34]. Interestingly, these studies were predominantly performed at late time points post viral infection in vitro when viral replication is established. We were able to show a significant upregulation of LDs in primary astrocytes infected with ZIKV and DENV (members of the *Flaviridae* virus family) at 8 h post-infection but could also see an observable down regulation of LDs at 2–3 days post infection of both viruses (Supplementary Fig. 9), indicating that it is not a cell type specific response, but rather a function of viral replication at later time points. To better examine the induction of LDs in the absence of viral antagonism of the early innate immune response, we analysed LD dynamics in response to synthetic dsRNA and dsDNA viral mimics (Fig. 3) where it was clearly observed that these PAMPs were able to elicit a rapid upregulation of LDs as early as 2 h post transfection, which peaked at around 8 h, and returned to baseline by 72 h post stimulation. This in part corresponds to what Barletta et al. (2016) demonstrated in their mosquito model, where LD accumulation was mimicked via synthetic activation of the Toll and IMD antiviral innate pathways[15], leading to the hypothesis that the accumulation of LDs may be an important antiviral response in the mosquito. It is interesting to note, that the number, size and composition of LDs vary greatly within cells in a homogenous population as well as in different cell types[35] and although all 5 cell types examined in this study were able to induce LDs upon activation of these pathways, the degree in which they could achieve this differed (Fig. 2b). Furthermore, the average size of LDs in different cell types was also shown to increase with the exception of LDs from THP-1 macrophages (a cell type that already displays a large average size of LDs without prior stimulation), perhaps demonstrating that there is an optimal size range for LDs in respect to their functional importance following a viral infection. Alternatively, as the LD sizes calculated are an average, it is still plausible that resident LDs in THP-1 macrophages have grown in size following stimulation, however a larger influx in nascent LDs may have reduced the overall average size for this particular cell type.

Astrocytes are well known for their fast type I interferon response which can be protective from flavivirus infection and virus-induced cytopathic effects[19,36]. Astrocytes also have a very robust type-III IFN response which contributes to their ability to be refractory to HSV-1 infection[37,38]. We were able to demonstrate that LD induction correlated with the production of both type I and III IFN, and that when impeded it significantly

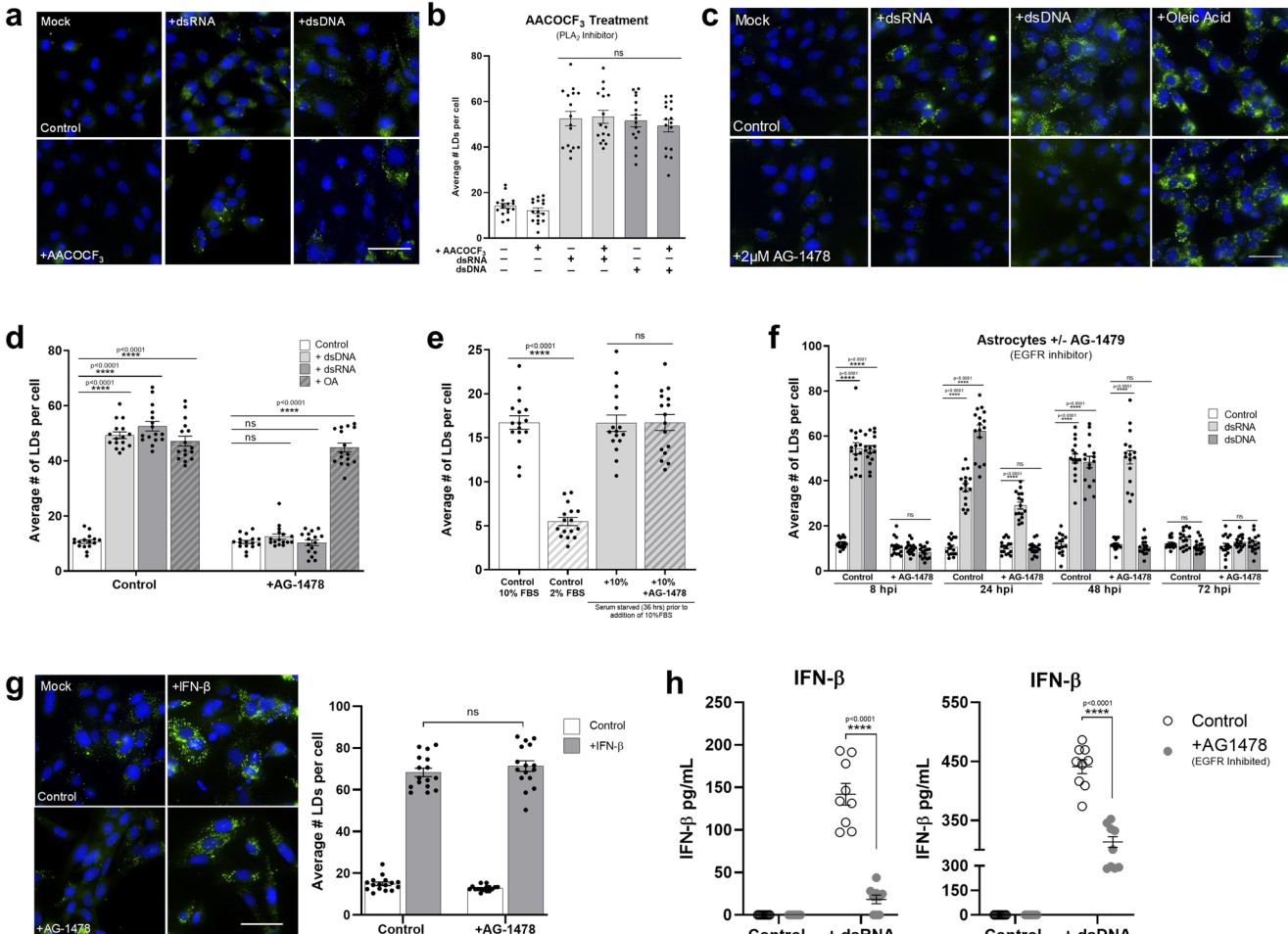

**Fig. 7 LD Induction following nucleic acid detection is EGFR mediated.** Primary immortalised astrocyte cells were treated with 2 μM AACOCF$_3$ (PLA$_2$ inhibitor) for 16 h prior to stimulation with dsRNA or dsDNA for 8 h and (**a**) were stained with Bodipy (493/503) to visualise LDs (green) and DAPI to visualise the cell nuclei (blue) 8 h post stimulation and (**b**) average numbers of LDs per cell was analysed using ImageJ analysis software (greater 200 cells, $n = 2$). **c** Primary immortalised astrocyte cells were treated with 2 μM AG-1478 (EGFR inhibitor) 16 h prior to stimulation with dsRNA or dsDNA, or treatment with OA and were stained with Bodipy (493/503) to visualise LDs and DAPI to visualise the cell nuclei 8 h post-stimulation. **d** The average number of LDs per cell analysed using ImageJ analysis software (greater 200 cells, $n = 2$). **e** Primary immortalised astrocyte cells were serum starved for 48 h, plated into wells and treated with 2 μM AG-14789 or control for 16 h. All cells were then given fresh full serum media for 36 h and stained to visualise LDs (green) as above. LDs were analysed using ImageJ analysis software. **f** Primary immortalised astrocyte cells were treated with 2 μM AG-1478 (EGFR inhibitor) for 16 h prior to stimulation with dsRNA and dsDNA for up to 72 h and were fixed at regular time points until 72 h post stimulation. Average numbers of LDs per cell was analysed using ImageJ analysis software (greater than 200 cells, $n = 2$). **g** Primary immortalised astrocyte cells were treated with 2 μM AG-1478 (EGFR inhibitor) for 16 h prior to stimulation with IFN- β and their LDs were assessed using image J analysis software. In (**b**–**g**) error bars, mean values ± SEM, $P$ values were determined by unpaired two-tailed Student's $t$ test with a Holm-Sidak correction for multiple comparisons for 2 or more groups (greater than 300 cells; $n = 3$ biological replicates). Stimulated cells were statistically compared with their respective mock controls, ns = not significant. **h** Primary immortalised astrocyte cells were treated with 2 μM AACOCF$_3$ for 16 h prior to stimulation with dsRNA or dsDNA for 16 h where secreted IFN protein levels from these experiments were analysed via ELISA for IFN- β and IFN-λ protein. Error bars, mean values ± SEM, $P$ values were determined by two-way ANOVA post-hoc pairwise comparisons with Bonferroni correction ($n = 3$ biological replicates). Source data are provided as a Source Data file.

impacted the transcriptional IFN response in these cells. In addition, when cellular LD numbers were enhanced in vitro, cells produced significantly higher secreted levels of both type I and III IFNs, which coincided with a significant drop in viral load in the infected cells. Together this suggests that the initial production of LDs following viral infection may play a significant role in limiting early viral replication, perhaps through an enhanced antiviral state in the cell. Interestingly, we were also able to demonstrate that dsRNA, and not dsDNA driven LDs were induced in a bi-phasic manner (Fig. 3), with the second wave likely being induced in an autocrine or paracrine manner following IFN secretion.

LDs are known to be induced via multiple mechanisms, with common LD biogenesis involving the accumulation of neutral lipids (most commonly TG and sterol esters) between the bilayers of the ER membrane, leading to the budding off of nascent LDs into the cytoplasm[39,40]. Several proteins are involved in LD biogenesis in mammalian cells, including PLA$_2$, perilipins (PLINs), triacylglycerol (TAG) biosynthetic enzymes, fat-inducing transmembrane proteins (FIT1 and FIT2), SEIPIN and fat-specific storage protein 27 (FSP27)[41] as well as some evidence of additional proteins involved in membrane dynamics (coatomer protein 1, SNAREs, Rabs and atlastin)[42]. Here we demonstrate that virally induced LDs have a different biogenesis

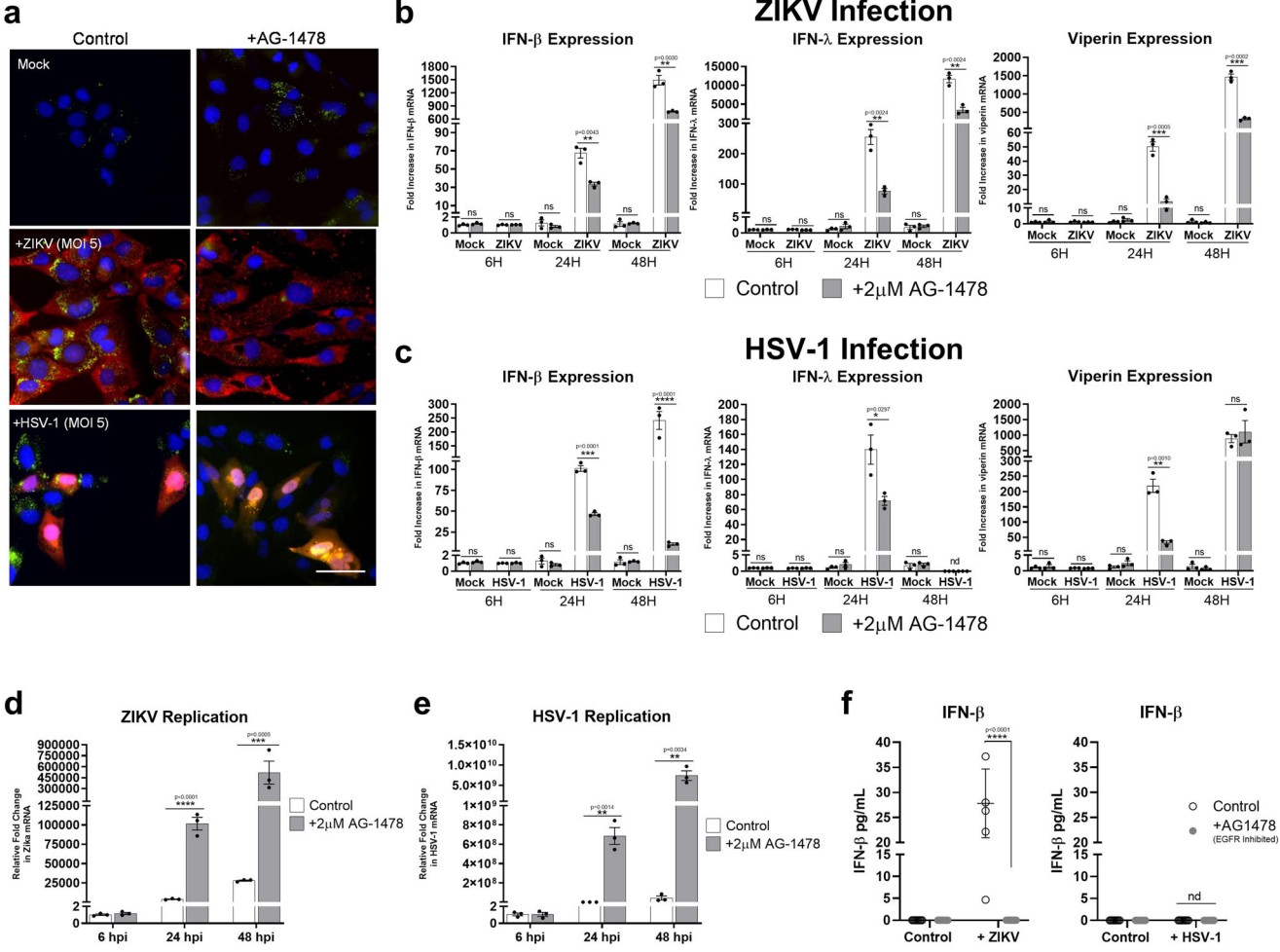

**Fig. 8 EGFR treatment enhances viral infection and dampens the interferon response to ZIKV and HSV-1. a** Primary immortalised astrocyte cells were treated with 2 μM AG-1478 (EGFR inhibitor) for 16 h prior to infection with ZIKV and HSV-1. Cells were stained with Bodipy (493/503) to visualise LDs (green) and DAPI to visualise the cell nuclei (blue). ZIKV RNA was detected using anti-3G1.1 and 2G4 dsRNA antibodies and HSV-1 was detected using the anti-HSV-1 antibody (Abcam, ab9533), both viral proteins shown with red staining. Images are a representation of $n = 3$ independent experiments. **b, c** RT-qPCR was utilised to evaluate IFN-β, IFN-λ and viperin mRNA expression at 8, 24 and 48 hpi for both ZIKV at MOI 0.1 or HSV-1 at MOI 0.01. RT-PCRs were performed to detect viral nucleic acid levels of (**d**) ZIKV and (**e**) HSV-1. In (**b–e**) error bars, mean values ± SEM, P values were determined by unpaired two-tailed Student's t test with a Holm-Sidak correction for multiple comparisons for 2 or more groups (greater than 300 cells; $n = 3$ biological replicates). Stimulated cells were statistically compared with their respective mock controls, ns = not significant. **f** IFN protein levels from these experiments were analysed via ELISA for IFN- β and IFN-λ protein at 16 h post infection. Error bars, mean values ± SEM, P values were determined by two-way ANOVA post-hoc pairwise comparisons with Bonferroni correction ($n = 3$ biological replicates), nd = not detected. Source data are provided as a Source Data file.

mechanism to the normal homoeostatic LD biogenesis, and that their production was driven independently of type-I IFN, however, both type-I and -III IFNs were able to stimulate the induction of LDs in astrocyte cells (Fig. 6). There have been previous reports of type- II IFNs (IFN-γ) inducing LDs during a Mycobacterium infection[13], however, to our knowledge there have been no reports of other interferon species activating LD upregulation. Interestingly, we found that both EGFR and PI3K, but not PLA₂, were driving the induction of LDs following viral infection, however this was not the case for LDs induced by IFNs (Fig. 7, Supplementary Fig. 7 and Fig. 9). EGFR has also previously been shown to elevate LD numbers in human colon cancer cells[27]. In addition, increases in LDs were blocked by inhibition of PI3K/mTOR pathways, supporting their dependency on selected upstream pathways. This fits with our findings that EGFR engagement plays a role in the induction of virally induced LDs. As mentioned above, we also observed a bi-phasic induction of LDs following dsRNA stimulation, which was firstly mediated by EGFR, in an interferon independent mechanism,

with a second wave of LDs being IFN inducible (Fig. 6). Blocking of IFNAR1 was able to significantly inhibit the second wave of LD upregulation but not completely abolish it, perhaps indicating that type-III IFNs may also play a role in this second wave of induction (Fig. 6f). It is interesting that this phenomenon was not observed following stimulation of cells with dsDNA, potentially indicating slightly different biogenesis pathways, or alternately the co-induction of a negative regulator of LD biogenesis (Fig. 9). Previous seemingly contradictory work has identified both an inhibitory and stimulatory role for EGFR in type-I IFN production[43–45].

We have shown that the upregulation of LDs following a viral stimulus plays an antiviral role in the cell; and our work has demonstrated that this upregulation contributes to a heightened type I and III interferon response in vitro. However, the exact mechanisms involved in this heightened antiviral state still remain to be elucidated. One possibility is that the LD is being utilised as a platform for protein sequestration that contributes to an enhanced IFN response. Previous work from our team has

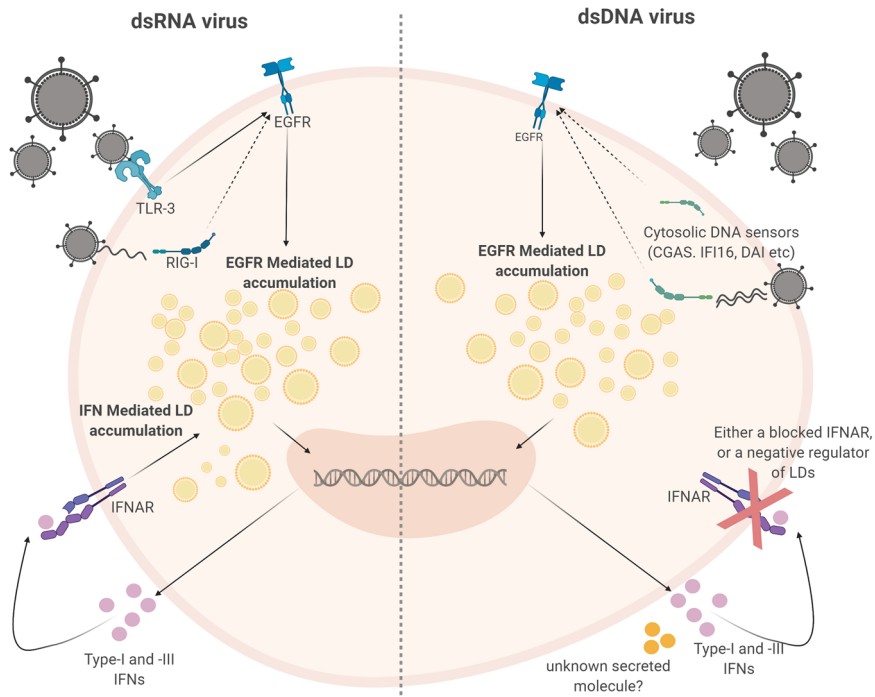

**Fig. 9 Lipid droplets are induced to enhance an effective interferon response.** Here we demonstrate that the host upregulates LDs following pathogen detection in two waves. The first wave is EGFR dependent which is independent of interferon production, and the second wave is induced via interferon production. During dsDNA stimulation, there is no second wave, and this is hypothesised to be due to a negative regulator of LD accumulation produced by cells during this response. Figure created by authors on BioRender.com.

extensively described the host protein, viperin as having both broad and specific anti-viral properties, which are largely dependent on its localisation to the LD[46–49]. Viperin's presence on the LD has been shown to significantly enhance the production of type I IFN following engagement of dsDNA receptors, as well as the TLR7/9 receptors[49,50]. In addition, a very recent proteomic study of LDs taken from LPS stimulated mice has revealed that several interferon-inducible proteins such as viperin, IGTP, IIGP1, TGTP1, and IFI47 were upregulated on LDs following this stimulation[51]. To-date there has been no studies describing the potential antiviral importance of the LD proteome following viral infection, and it is plausible that there may still be undiscovered antiviral effectors that require LD localisation.

There is an expanding appreciation for the roles of lipids in the antiviral response during infection, in particular, how they can contribute to the inhibition of viral infections. Lipids have been shown to play numerous roles in activation and regulation of immune cells such as T lymphocytes and macrophages[52]. Recently, a mechanism was described for the activation of macrophages through the release of a distinct class of extracellular vesicles, which are loaded with fat derived directly from adipocyte LDs[53]. As well as having a signalling role in activating immune cells, certain species of lipids have been shown to modulate immune responses. Polyunsaturated fatty acids (PUFAs) are precursors for the synthesis of numerous bioactive lipid mediators, such as eicosanoids and specialized pro-resolving mediators which are released from various immune cell types to modulate immune responses[54–56]. The PUFA lipid mediator D1 (PD1) has also been demonstrated to inhibit IAV infection in cultured cells[57]. It is also important to note that LD populations both between cells and within a cell are diverse, and can consist of different sizes, numbers and distinct protein or lipid compositions. However, the reason for LD diversity is still unclear[35,58–60]. Lipidomics is a growing field and could be utilised to investigate the role and composition of specific subsets of LDs within cells

both prior to, during and following viral infection, to give further insight into whether changes within the lipidome assist in driving an antiviral response[57].

The early induction of LDs following a viral infection acts to aid the antiviral host response by enhancing the production of interferon. Multiple viruses have been demonstrated to usurp host cell LDs to facilitate their replication cycles, and it is possible that this may also represent a subversion mechanism to disrupt early antiviral signalling, however further work is required to unravel these intersections. LDs are now considered a highly dynamic organelle involved in the facilitation of multiple cellular pathways and responses, and it is now clear that they are also involved in a pro-host response to viral infection.

## Methods
**Cells and culture conditions**. All mammalian cell lines were maintained at 37 °C in a 5% $CO_2$ air atmosphere. Huh-7 human hepatoma cells, HeLa human epithelial cells, HEK293T human embryonic kidney cells, primary murine embryonic fibroblast (MEF) cells, Vero cells, a green monkey kidney cell line, and Primary Immortalised Astrocytes were all maintained in DMEM (Gibco) containing 10% foetal bovine serum (FBS), 100 units/mL penicillin and 100 μg/mL streptomycin. Human monocytic cells (THP-1) were cultured in high glucose RPMI 1640 medium, supplemented with 10% FBS, 100 units/mL penicillin and 100 μg/mL streptomycin. C6/36 Aedes albopictus cells were maintained in Basal Medium Eagle (BME) supplemented with L-glutamine, MEM non-essential amino acids, sodium pyruvate, 10% FBS and P/S and cultured at 28 °C with 5% $CO_2$. For serum replacement experiments, cells were cultured in serum replacement 3 (Sigma, S 2640) in DMEM at a concentration of 10% prior to experiments. All experiments were then performed in serum replacement rather than DMEM + FBS.

**Influenza infection of mice**. C57BL/6 mice were bred in-house and housed under specific pathogen–free conditions in the animal facility at the Peter Doherty Institute of Infection and Immunity, University of Melbourne, Melbourne, Australia. Animal experiments were performed in accordance with the University of Melbourne's Animal Welfare Committee (1714189) and Institutional Biosafety Committee approval NLRD (2018/023); mice were maintained under a 12-h light/ dark cycle at an ambient temperature of 20–23 °C and relative humidity of 40–60% and with ample food and water. All experiments were done in accordance with the Institutional Animal Care and Use Committee guidelines of the University of

Melbourne. Mice were anesthetized with isoflurane and intranasally infected in a volume of 30 μL with $10^4$ plaque forming units (PFU) of mouse adapted influenza A viruses, x31(H3N2) or PR8(H1N1). Mock infected mice received 30 μL of PBS intranasally.

**Dengue infection of mice**. BALB/c mice were bred in-house and housed under specific pathogen-free conditions in the animal facility at Flinders University, Adelaide, Australia, mice were maintained under a 12-h light/dark cycle at an ambient temperature of 20–23 °C and relative humidity of 40–60% and with ample food and water. Animal procedures were performed in accordance with Flinders University Animal Welfare Committee approval number 935-17 and Institutional Biosafety Committee approval NLRD 2013-24. Near-term pregnant BABL/c mice were transferred to individually ventilated cages and monitored for delivery. At day 1 postpartum, mothers were removed from the cages and 1-day old BALB/c pups were infected by injection into the body next to the milk sac, with 800 PFU DENV-2 MON601 diluted in 1x PBS in a volume of 10 μL with an insulin syringe. Mock-infected animals (n = 3) were similarly injected with 10 μl 1x PBS. Pups were monitored daily for movement, skin colour and social inclusion. At day 2 or day 4 post-infection the mothers were rapidly euthanised by $CO_2$ asphyxiation and pups were euthanized by humane decapitation.

**In vitro viral infection and viral mimics**. Monocytes were seeded at $1 × 10^6$ per well in 12-well plates and pre-treated into polarisation states 24 h prior to infection with Influenza A Virus (IAV). Primary Immortalised Astrocyte cells were seeded at $7 × 10^4$ per well in 12-well plates prior to infection with Herpes Simplex Virus-1 (HSV-1), zika (ZIKV) or dengue (DENV) viruses. ZIKV (MR766 strain) DENV and HSV-1 (KOS strain) were diluted in serum-free RPMI at a MOI of 0.1. Cells were washed once with PBS then infected with virus. IAV strains PR8 (H1N1) and X-31 (H3N2) were diluted in serum-free RPMI to a MOI of 1.0. THP-1 monocyte cells were seeded at $1–3 × 10^6$ and were co-incubated with either PR8 or X-31 for 1 h in 200 μL AIM medium (RPMI-1640 medium supplemented with HCl to pH 6.0), followed by 8 h in 2 mL complete medium, RPMI-1640 medium supplemented with 10% foetal calf serum, at 37 °C containing 5% $CO_2$.

The viral mimics, poly dA: dT (dsDNA) and poly I: C (dsRNA) (Invivogen) were transfected into cells using PEI transfection reagent (Sigma-Aldrich, MO, USA) as per manufacturer's instructions at a concentration of 1 μg/ml. For interferon stimulations, 1000 U/mL IFN-β (PBL Assays) and 100 ng/mL IFN-λ (IL-29) (R&D Systems) were incubated on cells for 16 h (unless otherwise indicated).

**Primary murine astrocyte cultures**. The establishment of astrocytic cultures from the brains of C57BL/6 mice (post-natal day 1.5) was performed as described previously[61]. C57BL/6 mice were bred in-house and housed under specific pathogen-free conditions in the animal facility at La Trobe University; mice were maintained under a 12-h light/dark cycle at an ambient temperature of 20–23 °C and relative humidity of 40–60% and with ample food and water. Experiments were carried out under the approval of the La Trobe University Animal Ethics Committee (AEC 18-05) in accordance with Guidelines for Ethical Conduct in the Care and Use of Animals. Forebrains were dissected in ice-cold Hanks balanced salt solution (137 mM NaCl, 5.37 mM KCl, 4.1 mM $NaHCO_3$, 0.44 mM $KH_2PO_4$, 0.13 mM $Na_2HPO_4$, 10 mM HEPES, 1 mM sodium pyruvate, 13 mM d(+)glucose, 0.01 g·L$^{-1}$ phenol red), containing 3 mg·mL−1 BSA and 1.2 mM $MgSO_4$, pH 7.4. Cells were chemically and mechanically dissociated, centrifuged, and the pellet resuspended in astrocytic medium [AM: DMEM, Dulbecco's modified eagle medium, 10% FBS, 100 U·mL$^{-1}$ penicillin/streptomycin, 0.25% (v·V-1) Fungi-zone], preheated to 36.5 °C at a volume of 5 mL per brain and plated at 10 mL per T-75 cm² flask. Cells were maintained in a humidified incubator supplied with 5% $CO_2$ at 36.5 °C and complete medium changes were carried out twice weekly. When a confluent layer had formed (~10 days in vitro), the cells were shaken overnight (180 rpm) and rinsed in fresh medium to remove non-astrocytic cells. Astrocytes were subsequently detached using 5 mM EDTA (10 min at 37 °C), plated onto coverslips in 24-well plates at $1 × 10^4$ cells per well, and incubated in a humidified atmosphere at 36.5 °C with 5% $CO_2$ overnight. A full medium change was performed to remove non-adherent cells and medium was subsequently changed every 3–4 days thereafter until cells were ready for use.

**Lipid droplet induction and treatments**. Vehicle controls were used in all experiments performed with inhibitors or treatments and cell viability was monitored between vehicle controls and cells with no treatments.

*For enhancing lipid droplets*. Oleic acid (n-9 MUFA, C18:1)—a Long-chain fatty acid was used to increase LDs within cells. OA was purchased from Sigma (Sigma-Aldrich, MO, USA) and dissolved in 0.1% NaOH and 10% bovine serum albumin (BSA). OA was prepared as a 10 mM stock solution and stored at −20 °C. BSA was used as a vehicle control. Cells were treated with 500 μM OA in DMEM (+1%BSA) for 16 h.

*For serum starvation of cells*. Cells were either given low serum media containing 2% FCS, or control serum media containing 10% FCS and were incubated in T-75 cm² flasks for 48 h prior to plating at the required cell density. Cell culture media

on all experiments was changed 30 min prior to the beginning of the experiment, with all transfections and experiments being performed in 10% FCS.

*Inhibition of EGFR*. Tyrphostin AG-1478 (4-(3-chloroanilino-6, 7-dimethoxyquinazoline) mesylate, Mr 411.1) was manufactured by the Institute of Drug Technology (IDT, Melbourne, Australia) and solubilized in DMSO (stock 50 mM). Cells were grown in media containing 2 μM AG-1478 or an equivalent amount of vehicle (DMSO, 1:25,000 v/v). In all experiments AG-1478 media was discarded, and the cells were washed twice with 1x PBS before being followed in pre-warmed media without AG-1478 1 h prior to infection/stimulation.

*Inhibition of $PLA_2$*. $AACOCF_3$ (Abcam; ab120350) was utilised to inhibit $PLA_2$. $AACOCF_3$ was prepared in DMSO and stored at −20 °C. Aliquots were diluted in complete DMEM to 2 μM immediately prior to use. The final DMSO concentration was always lower than 0.1% and had no effect on LD numbers. DMSO was used as a vehicle control.

*Inhibition of PI3K*. Wortmannin is a well-described inhibitor of PI3K[62] and was obtained from Sigma, dissolved in DMSO at a concentration of 1 mM. Cells were grown in media containing 100 μM Wortmannin. In all experiments, Wortmannin media was discarded, and the cells were washed twice with PBS before addition of pre-warmed media without Wortmannin 1 h prior to infection/stimulation. DMSO was used as a vehicle control.

*Inhibition of DGAT1 and DGAT2*. The DGAT1 inhibitor, T863 and the DGAT2 inhibitor, PF-06424439 were used to inhibit the DGAT enzymes. To inhibit DGAT1, T863 was dissolved in DMSO at a concentration of 15 mg/mL. Cells were treated in media containing 2 μM T863 for 1 h prior to experiments. To inhibit DGAT2, PF-06424439 was dissolved in $dH_2O$ at a concentration of 10 mg/mL. Cells were treated in media containing 10 μM PF-06424439 for 1 h prior to experiments. In all experiments, DGAT media containing inhibitors was discarded, and the cells were washed twice with 1x PBS before the addition of infections/stimulations. DMSO was used as a vehicle control.

*Inhibition of IFNAR1*. IFNAR1 was inhibited via treatment with the MAR1 blocking antibody which has been well described[63]. Primary immortalised astrocyte cells were treated with 2 μM of either MAR1 or control blocking antibodies in complete media for 1 h prior to experiments. In all experiments, media containing antibodies was discarded, and the cells were washed twice with 1x PBS before the addition of infections/stimulations.

**IFN ELISAs**. Cell culture supernatant was analysed for IFN-β and IFN-λ release using commercial ELISA kits (Crux Biolab, Human IFN-beta ELISA kit (EK-0041) and RayBiotech inc., Human IL-29 ELISA (ELH-IL29-1)) following the manufacturer's instructions. ELISA plates were read at an absorbance of 450 nm using a CLARIOstar microplate reader and MARS software (BMG LabTech, Ortenberg, Germany).

**Conditioned IFN media experiments**. Primary immortalised astrocyte cells or Vero cells were stimulated with dsRNA and dsDNA viral mimics for 4 h before being washed and replenished with fresh complete DMEM media and left to produce IFNs for a further 12 h. Media was then taken from these cells, centrifuged to remove any cell debris and placed on freshly seeded unstimulated cells. These cells were left in this conditioned media for 8 h and fixed with 4% paraformaldehyde (PFA) and their LD numbers were analysed.

**Immunofluorescence microscopy**. For cultured cells, briefly, cells were grown in 24-well plates on 12 mm glass coverslips coated with gelatine (0.2% [v/v]) were washed with PBS, fixed with 4% paraformaldehyde in PBS for 15 min at room temperature and permeabilised with 0.1% Triton X-100 in PBS for 10 min. Cells were blocked with 1% BSA for 30 mins, before antibody staining with anti-3G1.1 and 2G4 dsRNA antibodies (for ZIKV, DENV) or anti-HSV-1 (1:200). Cells were then incubated with Alexa Fluor 555 secondary antibody (1:200) for 1 h. LDs were stained by incubating cells with Bodipy (493/503) at 1 ng/mL for 1 h and nuclei were stained with DAPI (Sigma-Aldrich, 1 μg/ml) for 5 min at room temperature. Samples were then washed with PBS and mounted with Vectashield Antifade Mounting Medium (Vector Laboratories). Preparation and staining of murine lung and murine head frozen sections were prepared following optimised protocols for tissue sections[64]. First, frozen lung sections were prepared by inflating the lungs with optimum cutting temperature (OCT). Whole heads were sectioned sagittally. One section was snap-frozen and the other mounted immediately in OCT. Frozen sections were cut at 14 μM with a Leica CM 3050 S cryostat and mounted on microscope slides and stored at −80 °C. Sections were fixed with 4% paraformaldehyde in PBS for 15 min at room temperature. Sections were then washed with PBS, permeabilised with 0.1% Triton X-100 in PBS for 10 min, washed again and then blocked with 1% BSA for 30 mins. Sections were incubated with either 1:1000 αIAV NP (for lung sections) or neat anti-3G1.1 and 2G4 dsRNA antibodies (for head sections) for 1 h. Sections were then washed and incubated with Alexa

Fluor 555 secondary antibody at 1:200 for 1 h. Bodipy (493/503) was used to stain for LDs at 1 ng/mL for 1 h at room temperature, and nuclei were stained with DAPI for 5 min at room temperature. Images were then acquired using either a Nikon T*i*E inverted fluorescence microscope or Zeiss confocal microscope. Unless otherwise indicated images were processed using NIS Elements AR v.3.22. (Nikon) and ImageJ analysis software.

**Lipid droplet enumeration**. LD numbers and diameters were analysed using quantitative data from the single raw ND2 images (from NIS elements) in ImageJ using the particle analysis tool. For each condition, at least 9 fields of view were imaged at 60X magnification from different locations across each coverslip. LDs from at least 100 cells per biological replicate with a minimum of $n = 2$ per experiment being analysed for both LD number and average LD size.

**RNA extraction and real time PCR**. All experiments involving real-time PCR were performed in 12-well plates with cells seeded at $1 \times 10^6$/well (monocytes and macrophages) or $7 \times 10^4$/well (all other cell types) 24 h prior to infections/stimulations and performed at least in triplicate. Total RNA was extracted from cells using TriSure reagent (Bioline), with first strand cDNA being synthesized from total RNA and reverse transcribed using a Tetro cDNA synthesis kit (Bioline). Quantitative real-time PCR was performed in a CFX Connect Real-Time Detection System (BioRad) to quantitate the relative levels of IFN, and interferon stimulated gene mRNA in comparison to the housekeeping gene RPLPO. Primers sequences can be found in Supplementary Table 1.

**Statistical analysis and reproducibility**. Results are expressed as mean ± SEM. Student's t tests were used for statistical analysis between 2 groups, with $p < 0.05$ considered to be significant. Experiments with 2 or more experimental groups were statistically analysed using an ordinary two-way ANOVA with multiple comparisons. All statistical analysis was performed using Prism 8 (GraphPad Software). All experiments were performed in biological triplicate (unless otherwise stated), and technical duplicates for RT-PCRs. All image analysis were derived from 16 individual field of view images with over 200 individual cells.

**Reporting summary**. Further information on research design is available in the Nature Research Reporting Summary linked to this article.

## Data availability
The data that support this study are available within the article and its Supplementary Information files or available from the authors upon request. Source data are provided with this paper.

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

## Acknowledgements

This work was funded by a La Trobe University Research Focus Area grant, as well as a NHMRC ideas grant (APP1181434) to K.J.H. and D.R.W.. L.W. is funded by an ARC Future Fellowship, D.W. is funded by an ARC DECRA. The authors would like to acknowledge Peter Lock from the La Trobe University Bioimaging Platform, Paul Hertzog from the Centre for Innate Immunity and Infectious Diseases, Hudson Institute of Medical Research, for provision of IFNAR antibodies and Roy Hall from the Australian Infectious Disease Research Centre at the University of Queensland for gifting the dsRNA antibodies used throughout this manuscript.

## Author contributions

E.A.M. performed the majority of the experiments; M.D. and W.C. assisted in vitro influenza studies, and L.W. and J.C. performed the murine in vivo studies. R.O. assisted in the isolation of murine astrocytes, and K.M.C. assisted in experiments involving MEFs. K.J.H. was responsible for the overall study design, with E.A.M. and D.R.W. also assisting in experimental direction. K.J.H. and E.A.M. wrote the manuscript; all authors commented on the manuscript.

## Competing interests

The authors declare no competing interests.
