## [Peer Review File · Nature Communications]

Reviewer comments, first round -

Reviewer #1 (Remarks to the Author):

In this well-written manuscript, Monson et al. follow up on their previous observation that loss of lipid droplets (LDs) impairs antiviral immune response and attempt to identify the associated mechanisms. The authors suggest that LD induction corresponds to type I and III IFN production which results in viral clearance thus, suggesting an important role of LDs in antiviral immunity. Past publications have demonstrated the importance of LDs in viral assembly and how viruses manipulate these organelles to their advantage. In the present manuscript, the authors argue that while viruses may utilize LDs during later stages of infection, the LDs induced initially during viral infection contribute to IFN-mediated antiviral response. The authors also determine EGFR engagement as an important factor for LD induction after viral infection. Although this manuscript addresses a fairly novel role of LDs in antiviral immunity, overinterpretation of the data obtained, lack of appropriate control experiments and not addressing the shortcomings of the studies in the discussion reduce the enthusiasm for this manuscript.

This reviewer hopes that following specific comments will help the authors strengthen their findings:

Major revisions:

1. Cellular model and viruses used:

The authors fail to explain the rationale behind the particular viruses and the corresponding cellular models used. There is minimal existing data on the role of LDs during ZIKV and HSV infection. Since the authors mention that existing data for other viruses utilizing LDs for assembly is during later stages of infection, using HCV or DENV to demonstrate the LD-mediate antiviral immunity early on during infection would be more convincing.

The authors also use astrocytes for majority of their studies without a convincing rationale in the text. Also, the rationale behind using Influenza A virus only in Fig 1 and the only virus for in vivo studies is unclear.

2. Specificity of the phenotype:

Since the previous Monson et al. publication demonstrating that LDs are important for host response against dsRNA uses HeLa and Huh-7 cells, it seems like activation of any cell type using dsRNA will increase LDs and thus the LD-mediated antiviral activity. This effect seems highly non-specific and suggests this could be simply a cellular response to any antigen. Determining how these cells react to non-viral ligands like LPS, lipoproteins etc. might provide a better insight to if the effect is virus specific or a cellular response to any stimulation.

3. The studies that the authors have performed to mention that LDs are responsible for IFN-mediated antiviral immunity are not conclusive. The direct relationship between LDs and type I IFN can be established by blocking LD formation and/or breakdown. This could be either achieved with inhibitors or knockout cells. This is a very important piece of data that is missing from the manuscript and is essential to establish the correlation between LDs and IFN in virus-infected cells.

4. The authors elude that viruses utilize LDs eventually and demonstrate this by time-dependent reduction of LDs in infected cells. However, to accurately interpret eventual usage of LDs, the authors should perform a LD turnover assay and determine if in fact viruses are utilizing them.

5. There are several other aspects that have not been discussed in the manuscript – which receptors are the dsDNA and dsRNA functioning through? Why does dsRNA induce increased IFN levels compared to dsDNA?

6. Figure-specific comments:

Figure 1: LD accumulation in response to different viruses

- In Fig 1 and Fig S1, why do the astrocytes look different between zika-infected and HSV-infected panels?
- In Fig S1, what do the white boxes in mock, ZIKV and mock, HSV infection represent?
- In Fig 1B and C, the nucleus in mock infection looks larger than the infected cells. Is it at a different scale? Why is only one cell shown? It would be ideal to see a larger field at a lower magnification and then higher magnification of an inset.
- In Fig 1A-C, is the red staining autofluorescence?
- Please clarify if the number of LDs were counted by imaging z-stacks. In the figures as is, it seems difficult to quantify. If z-stacks were not used, how were different LDs appropriately counted since they seem to overlap in the figures?
- Please discuss the importance of the LD size differences observed in Fig 1D.
- In this figure the authors mention that there are more basal level of LDs in astrocytes. In that case, please justify the use of oleic acid to stimulate more LD accumulation?

Figure 2: Treatment with dsRNA and dsDNA accumulates LDs

- The authors mention that nucleic acid stimulation drives both the generation of new LDs as well as growth of existing LDs. It is also possible that the existing LDs are breaking down during this process. Based on just differences in size, it is difficult to interpret the data. The authors could potentially address this by performing a fatty acid pulse-chase experiment as performed by Rambold et al., *Developmental Cell*, 2015.
- Fig 2D: Is this figure for dsRNA or dsDNA-treated cells?

Figure 3: LD accumulation is transient

1. Although the authors mention that dsRNA and dsDNA-mediated LD induction coincide with the IFN levels, at 24h dsRNA-treated cell have highest IFN levels whereas LD numbers are dropping down at this time point. Please write a more accurate interpretation.

Figure 4: Increased LD numbers enhance Type I and III IFN response to viral infection

1. Increase in IFN after dsRNA and dsDNA treatment could simply be a host response irrespective of LDs. As mentioned above, performing these studies in cells lacking LDs would answer this question.
2. What is the distribution of pattern recognition receptors on astrocytes and would that play a role in IFN production in response to viral antigen-stimulation?
3. In Fig 4D and E, if the IFN β mRNA level is not significantly different in dsDNA-treated cells, how do the authors explain the increase in IFN β protein levels after treatment with OA and dsDNA in Fig 4G compared to control?
4. In Fig 3, IFN β and lambda mRNA fold increase rises from 8h to 24 hours with 24hours being at its maximum. However, in Fig 4, especially for the dsRNA-treated cells, the mRNA fold increase is much lower at 24 h compared to 8h in the control. Why?

Figure 5:

1. The authors show IFN protein levels for ZIKV but not HSV? Why?
2. Why does ZIKV, an RNA virus, does not induce IFN protein production during infection of control cells but does in OA-treated cells? This is especially important to know since ZIKV is shown to induce LDs in Fig 1. This further weakens the argument that LDs induce IFN-mediated antiviral response.
3. Showing LDs numbers in ZIKV and HSV-infected cells after oleic acid treatment will help better interpretation of this figure.
4. Line 201-202 – The authors mention 'astrocytes with upregulated LDs'. However, Fig 5 does not give proof of that. LDs should be counted before concluding their role in IFN production.

Figure 6: Initial LD accumulation is IFN independent

1. To strengthen the IFN-independent initial induction of LDs, could the authors use IFN knockout astrocytes (from IRF7 KO mice) or any other feasible resources?
2. Based on Fig 6E, is IFN β the most important IFN since astrocyte dsRNA media-treated Vero cells show similar LD #s as that of 1000U/ml IFN β -treated cells? What is the role of IFN λ that is shown

to increase in previous figures?

3. Most experiments in this figure are only performed n=2 questioning the statistical significance of this data.

Figure 7: Role of EGFR in LD induction

1. Although the authors mention DMSO has no effect, this reviewer would like to see a vehicle control for Fig S6A,B.

2. If possible, please change Fig S6A. The image is blurry and difficult to observe differences.

3. Line 269-270 – Could you please clarify the precise role of EGFR?

Figure 8: EGFR-induced LDs impact IFN production and attenuate viral infection.

Could the authors please speculate in their discussion if LDs are induced early during viral infection resulting in IFN production which can attenuate viral replication, how are the viruses surviving during in vivo infections?

Inconsistencies in the paper:

- Could the authors explain the rationale behind different MOIs for ZIKV between figures? For example: Fig 1 – MOI – 5 for ZIKV and HSV, Materials – MOI - 0.1 for HSV and Zika.
- All figure legends lack several important details which are mentioned in materials and methods. Having all the details in the legend, especially times post-infection/treatment will help with analyzing the figures.

Statistical analyses:

- Although the analysis seems sound, the legend does not mention what the significance is compared to, leaving it up to the reader's interpretation.

Minor revisions:

Grammatical and writing errors:

1. Introduction:

- Line 52 – Italicize bacteria names
- Line 56 – mentions Trypanosoma cruzi infection of macrophages also induces LDs but takes 6-12 days to occur following infection. The authors should correct this sentence as reference#8 have measured LD numbers at 24 hours post-infection in cells including peritoneal macrophages.
- Line 67-68 – Italicize M. tuberculosis
- In the introduction, the authors do not mention HCV whose relation with LDs is extensively studied.
- Line 122 is quite confusing. Do the authors mean after stimulation or in unstimulated cells?
- Line 124-125 mentions THP-1 average basal LD size did not increase with stimulation. But Line 126-137 – dsRNA stimulation 2-fold increase in THP-1 monocytes? Please clarify.
- Line 173-174 – sentence is unclear. Are the authors talking about cells treated with dsDNA?
- Line 512 - mentions 'INF' instead of 'IFN' used throughout the manuscript. Please change for consistency.

Reviewer #2 (Remarks to the Author):

In this study Monson et al. describe the induction of lipid droplets (LDs) and Interferon response in relation to viral infection and its mimetics. The topic is highly interesting and the study provides informative data regarding lipid body induction kinetics. However, more mechanistic studies on LD induction in infection models are lacking, conclusions are drawn based on limited data sets and key experiments are missing. The experiments investigating the role of EGFR on LD induction are novel and appealing but lack mechanistic insights. IFN induction and LD induction upon viral infection is not novel, and the current data are insufficient to show causal relationships.

Major points

- The authors state that different types of LDs are present within their models. An additional

experiment showing that those LDs are actually different, based on LD marker expression or content would make this statement more convincing.

- In figure 1abc, a co-localization of the virus with LDs is seen. The effects described in the manuscript suggest a signaling role of LDs, especially on IFN response. However, if co-localization of LDs is seen with the virus, could this mean a direct virus-LD interaction?

- In line 145-149 the authors describe a bi-phasal induction of LDs upon dsRNA stimulation. This seems to be based on the drop of LD numbers in one time measurement (24h, figure 3b). A further time course including multiple time points is necessary to show this is an actual bi-phasal response. Moreover, in the discussion, line 381-383, the authors conclude that this bi-phasal response was mediated by EGFR and later by IFN. However, the experiments performed are not sufficient to approve or reject this hypothesis.

- In line 181- 183 the authors conclude that presence of LDs was able to enhance IFN production. However, experiments to show a causal relationship between LDs and IFN production are lacking. Therefore this conclusion cannot be drawn based on the experiments performed.

- The data of IFN production does not seem to be consistent between figures. In figure 4d and 4e is not consistent with the data from figure 3c and 3d. In figure 4 the expression of IFN is higher at 8h for dsDNA, while figure 3 shows that IFN is at 8h higher for dsRNA. The IFN of dsRNA goes down in time (from 8 to 24h hours) in figure 4, while it goes up in figure 3.

- Regarding figure 5, in line 200-202 the authors state that reduction in viral load for ZIKV coincided with enhanced levels of IFN and with upregulated LDs. However, no LD quantification is present for figure 5 neither for ZIKV or HSV-1 infection in this setting, making it impossible to prove that indeed LD numbers coincide with IFN production and reduced viral load in this setting. The LD quantification of figure 1 is performed with a higher MOI, making direct comparison impossible. This also counts for figure 8, where LD quantification is missing.

- In figure 5c and 5d, the virus is still able to replicate, even in the presence of IFN and LDs, and only seems to be inhibited by addition of Oleic Acid. In this experiment it is not proven that the presence of LDs themselves have an actual inhibiting effect on viral replication.

- In figure 5e, why is infection with the ZIKV not able to induce IFN production without Oleic Acid treatment? The authors have shown that mRNA and protein levels of IFN can differ, so the protein levels of IFN production upon HSV-1 infection should also be shown.

- In figure 6, in our opinion it is possible that the increase in LDs seen in Vero cells upon treatment with dsRNA astrocyte media could be induced by the presence of IFN in the media, as Vero still have an IFN α / β receptor. Therefore, the control treatment of Vero cells with IFN is missing.

- The conclusion in line 267-270 is not clear. Apparently, there are two mechanisms involved in the LD induction seen, one IFN mediated, and one EGFR mediated. This should be written more clearly.

- Data about the importance of the described processes in vivo is lacking and is overstated in the summary.

Minor points

- Bodipy 409/505 is used in the study. Please state the company, since the commonly used dye to stain LDs is Bodipy 493/503.

- In the summary, line 33-35, the authors state that LD induction is IFN independent. However, in the discussion they explain that the second part of the bi-phasal induction might be IFN dependent. Please be consistent.

- Please add bars when showing significance to allow the reader to follow which results were directly compared and significant.

- Please elaborate why these specific infection models (virus and cell type) were chosen for the current study.

- Additional information about cell viability would be interesting to add, since LDs can also be seen in stressed cells.

- Please add the labels of the antibody and color shown in the confocal pictures in the figure itself.

- In figure 1a the more virulent PR8 and less virulent X-31 strain of influenza are used. Could you elaborate why the X-31 seems to infect the cells better in this model (more cytoplasmic staining)?

- In figure 1b the ZIKV is stained using two antibodies, one of which for dsRNA. Could you please explain why, and also which staining is visible in which color and which picture since only three colors are seen in figure 1b instead of four.

- In line 94-95 it is mentioned that astrocytes have a high basal level of LDs (approximately 15 per cell). In the confocal pictures this is not clearly visible. Please add a clear confocal image showing

the basal level more clearly.

- In figure 1d a LD staining in lung tissue is shown, are those really LDs, since the staining seems not be inside cells? Please state what is stained in blue.
- Please describe if the LD count in virus infected samples is performed for all cells present in the sample or only for the infected cells within that sample.
- In line 22, the conclusion is not correct, since only for THP-1 cells the size of the LDs does not change, for the other cells it does.
- In Supplementary figure 2, please describe what is meant with "serum replacement medium". Also the sample names of the confocal pictures in Supplementary 6b is unclear and does not seem to match with the quantification. Is "replenished serum" with 2 or 10% FCS?
- In line 181- 183 the authors state a high amount of LD leads to IFN production. However, in figure 4, the combination of Oleic Acid with dsDNA does not lead to an additional increase in LD numbers compared to dsDNA alone, but it does lead to an increase in IFN production on protein level. How can this be explained?
- As the authors state in line 200-202 that reduction in viral load and enhanced IFN production coincide with LD number, it would be interesting to see the viral load of figure 5c and 5d at 72h as it is the time point that LD numbers are returning to basal levels again in the dsRNA/dsDNA setting.
- The graph of figure 7b has a mistake in the labeling of the last condition.

Reviewer #3 (Remarks to the Author):

In the manuscript by Monson et al. the authors study the impact of lipid droplet (LD) formation upon viral infection and show evidence that type I interferon (IFN) production relies on this process. They show data with a DNA virus (HSV-1) and an RNA virus (ZIKV), which highlights the importance of this pathway during viral infections.

The proposed mechanism is that as viruses infect cells, LDs are stimulated and IFN is produced by infected cells. The authors show that there is an increase in LDs and type I IFN in the presence of oleic acid in the context of infection. On the other hand, they show that pretreatment with an EGFR inhibitor (which blocks LD biogenesis) results on an increase in virus production and a decrease in LD production and IFN production. The authors have a previous publication showing the effects of oleic acid on LD formation and Sendai virus infection. In the present manuscript the authors expanded those studies to test also a DNA virus and use more relevant primary systems, such as astrocytes, although it is not fully elucidated.

Major comments:

1. Overall, the manuscript is well written, even though at times it is difficult to understand the rationale for the proposed experiments until later on. More details and background information at the beginning of some of the sections would be helpful (see below).

2. From Figure 3 onwards, the authors switched back and forth between infectious virus to DNA/RNA and then back to infectious virus. It is not well explained why do they do such change. The following questions arise:

Could direct DNA/RNA stimulation lead to different times of LD production compared to a real infection?

Is it possible that the outcome of LD production would be different in the context of cells dying due to infection?

Is the amount of DNA/RNA they use to stimulate comparable to the amounts that would be produced in a cell upon infection? Is it possible you are overstimulating?

3. Fig. 6E: It would be useful to quantify the amount of IFN they had in the conditioned media previous to addition to naive astrocytes. Would also help for understanding what is going on in the

DNA stimulated conditioned media. In that sense, maybe spiking the DNA stimulated conditioned media with IFN and having a "positive control" media with known amount of IFN could help too.

4. Fig. 7B: Why are they showing LD size instead of LD count? Can't really compare with following panels (7D, for instance). From the microscopy it looks that while there is LD production, it is not much as compared to observed in Fig 2A. Is it possible that indeed the AACOCF3 is inhibiting LD production as well in the presence of virus too? Why would the enzyme only inhibit PLA in the context of mock infection but not in the context of viral infection (assuming there is only way LD biogenesis pathway)? The control with oleic acid here would be a good addition, like in Figure 7C.

5. The in vivo studies are quite limited.

Minor comments:

1. On multiple occasions they use acronyms that were not properly introduced. Line 34, 77, 89, 242, 485

2. They have to homogenize criteria for presenting references, either "9,10" (line 60) or "5-7" (line 55).

3. Figure 4G: Could they specify exactly what are we seeing in panels F and G? They look exactly the same in the fig. Ideally, one should be able to follow by reading the manuscript and looking at the figures what are the authors trying to show. As it is, it's not possible to do so without going to the figure legends.

4. Figure 4 G and F and Fig. 5: Overall, it is not clearly explained/understood in the manuscript why do they do some of the experiments until later on. For instance, not very clear why do they treat with oleic acid? To over-stimulate production of LDs with what purpose? We know already that LDs are over produced in the presence of virus or dsDNA/dsRNA so why the need to add oleic acid? And we know there is IFN stimulation from presence of dsRNA/dsDNA or from LDs from literature or their own previous work. Then, when you reach in the text Figure 5, it becomes more evident they are trying to assess the antiviral effect of LDs.

5. Line 267 – Why did you chose the 16-hour time point? It would have been more informative to maintain the same ones.

6. Fig 7 - They are missing cell viability assays to show drugs at the concentration used are not killing the cells.

7. Line 521 – This should say size instead of number.

Reviewer #1 (Remarks to the Author):

In this well-written manuscript, Monson et al. follow up on their previous observation that loss of lipid droplets (LDs) impairs antiviral immune response and attempt to identify the associated mechanisms. The authors suggest that LD induction corresponds to type I and III IFN production which results in viral clearance thus, suggesting an important role of LDs in antiviral immunity. Past publications have demonstrated the importance of LDs in viral assembly and how viruses manipulate these organelles to their advantage. In the present manuscript, the authors argue that while viruses may utilize LDs during later stages of infection, the LDs induced initially during viral infection contribute to IFN-mediated antiviral response. The authors also determine EGFR engagement as an important factor for LD induction after viral infection.

Although this manuscript addresses a fairly novel role of LDs in antiviral immunity, overinterpretation of the data obtained, lack of appropriate control experiments and not addressing the shortcomings of the studies in the discussion reduce the enthusiasm for this manuscript.

This reviewer hopes that following specific comments will help the authors strengthen their findings:

Major revisions:

1. Cellular model and viruses used:

The authors fail to explain the rationale behind the particular viruses and the corresponding cellular models used. There is minimal existing data on the role of LDs during ZIKV and HSV infection. Since the authors mention that existing data for other viruses utilizing LDs for assembly is during later stages of infection, using HCV or DENV to demonstrate the LD-mediate antiviral immunity early on during infection would be more convincing.

The authors also use astrocytes for majority of their studies without a convincing rationale in the text. Also, the rationale behind using Influenza A virus only in Fig 1 and the only virus for in vivo studies is unclear.

Multiple cell types are utilised in the first 2 figures of this manuscript, however, to fully elucidate the mechanisms and dynamics of LD upregulation following viral infection, we chose to continue the work in the manuscript using astrocytes. The following text has now been added to the manuscript at line 153:

“Astrocyte cells were chosen to further examine the role of LDs in viral infection due to their extensive upregulation of LDs following viral stimulation and infection. Additionally, astrocytes are known to be rapid producers of an effective antiviral response (19).”

The choice of viruses was based on ensuring that we covered both an RNA (Influenza and ZIKV) virus, as well as a DNA virus (HSV-1). However, we thank the reviewer for the suggestion of including a flavivirus that is known to utilise lipid in its replication cycle and have now included both in vivo (Figure 1E and Sup-1B) and in vitro (Fig 1B, C and Sup-1A) analysis of upregulated LD regulation following DENV infection in mice and astrocytic cell lines.

2. Specificity of the phenotype:

Since the previous Monson et al. publication demonstrating that LDs are important for host response against dsRNA uses HeLa and Huh-7 cells, it seems like activation of any cell type using dsRNA will increase LDs and thus the LD-mediated antiviral activity. This effect seems highly non-specific and suggests this could be simply a cellular response to any antigen. Determining how these cells react to non-viral ligands like LPS, lipoproteins etc. might provide a better insight to if the effect is virus specific or a cellular response to any stimulation.

We thank the reviewer for their comment and agree that this may be a non-specific host response, however an important response that is able to drive heightened IFN production as outlined in this manuscript. In the introduction to this manuscript, we inform the readers that bacterial infections are known to upregulate LDs, and we believe that activation of PRRs will be the trigger for this, however this requires extensive research that is currently underway in our laboratory, but outside of the scope of this manuscript. As an example, seen below is a figure demonstrating that Huh-7 cells, which are absent in TLR3 and have very reduced RIG-I, do not upregulate LDs when stimulated with dsRNA, however, are able to upregulate LDs when treated with oleic acid.

In our previous paper, Monson et al, we are looking at LD reduction, and demonstrate that by reducing LDs in the two cell types mentioned, we reduce the IFN output and hence heighten viral replication; suggesting that the LD is tied to an effective antiviral innate immune response.

3. The studies that the authors have performed to mention that LDs are responsible for IFN-mediated antiviral immunity are not conclusive. The direct relationship between LDs and type I IFN can be established by blocking LD formation and/or breakdown. This could be either achieved with inhibitors or knockout cells. This is a very important piece of data that is missing from the manuscript and is essential to establish the correlation between LDs and IFN in virus-infected cells.

In figure 8 we already demonstrate that by blocking EGFR (which we show to be a requirement for virally induced LDs), we significantly reduce the production of type I and III IFN transcription. However, we now include an analysis of protein expression in figure 8F demonstrating that when we block EGFR and/or de novo TG synthesis, we see a much-reduced output of IFN.

4. The authors elude that viruses utilize LDs eventually and demonstrate this by time-dependent reduction of LDs in infected cells. However, to accurately interpret eventual usage of LDs, the authors should perform a LD turnover assay and determine if in fact viruses are utilizing them.

The ability of flaviviruses, such as DENV and ZIKV (both are now included in Supplementary figure 9) to usurp LDs and deplete them over time is a well-established phenomenon in the literature as referenced in the discussion already. We merely sought to establish that this was also the case in our astrocytic model of infection.

5. There are several other aspects that have not been discussed in the manuscript – which receptors are the dsDNA and dsRNA functioning through? Why does dsRNA induce increased IFN levels compared to dsDNA?

As mentioned in the answer above to question 2, we agree that the delineation of the exact PRRs involved in driving an increased LD response will be interesting, although receptors such as TLR4 and TLR2 have already been delineated for bacterial upregulation of LDs. However, we feel that this is outside of the scope of this manuscript, and that the universal involvement of EGFR in the upregulation of LDs from both DNA and RNA viral infection provides previously unknown mechanistic basis. Additionally, the viral mimics utilised in this manuscript are known to act through RIG-I and TLR3, as well as multiple dsDNA sensors as mentioned already at line 113, and below:

“we stimulated these cells with the synthetic viral mimics, dsRNA (poly I:C, known to mimic viral RNA pathogen associated molecular patterns (PAMPs), and activate the RNA sensors RIG-I and TLR3) or dsDNA (poly dA:dT, known to mimic DNA viral PAMPs, and activate cytosolic DNA sensors)”

In respect to the differential output of IFN following either dsRNA or dsDNA stimulation, we feel that this cannot truly be assessed using the time points in this manuscript, as it may simply be a time to response rather than a true difference; as evidenced by the fact that mRNA for type I and III IFN is often greater in the dsRNA treated cells, however protein output is higher in the dsDNA treated cells (Figure 4). However, we feel that the significant upregulation of LD induced IFN following both simulations is more important than the differential between the two simulations.

6. Figure-specific comments:

Figure 1: LD accumulation in response to different viruses

- **In Fig 1 and Fig S1, why do the astrocytes look different between zika-infected and HSV-infected panels?**

We apologise for this, the issue was in the magnification of the microscopy images, which have now been altered to be the same magnification.

- **In Fig S1, what do the white boxes in mock, ZIKV and mock, HSV infection represent?**

In figure S1, the white boxes represent the zoomed section for images in Fig 1. We apologize that this information was omitted from the figure legend, and we have now added this.

- **In Fig 1B and C, the nucleus in mock infection looks larger than the infected cells. Is it at a different scale? Why is only one cell shown? It would be ideal to see a larger field at a lower magnification and then higher magnification of an inset.**

The astrocyte cell line is a primary immortalised fetal cell line, and displays a heterogeneous size of cells, as can be seen in Fig S1A; this is why the nucleus appears larger in Figure 1B and 1C. However, we have now altered the individual cell that is shown to reflect a similar-sized magnification. We have elected to show individual cells in Figure 1, as an inset from Figure S1 (White box displays the inset shown in Figure 1). We feel that a larger magnification image better displays the LD upregulation in the cells following viral infection.

- In Fig 1A-C, is the red staining autofluorescence?

The red staining in Figures 1A-C is viral proteins from their respective viruses, as outlined in the figure legend. We apologise for not stating this clearly in the figure legend and have now changed the text to read that red staining is associated with viral protein presence in the cells.

- Please clarify if the number of LDs were counted by imaging z-stacks. In the figures as is, it seems difficult to quantify. If z-stacks were not used, how were different LDs appropriately counted since they seem to overlap in the figures?

The methods section of the manuscript has now been changed to reflect that single images were utilised. Prior to imaging analysis, we performed a data analysis to determine the differential between the use of imaging z-stacks to quantify LDs versus single ND2 raw images taken from NiS elements software. As can be seen in the attached figure, there was no significant difference, which we believe is due to the low profile of fixed cell culture cells, hence single images were utilised for ease of use. Additionally, LD images for quantification were collected on a Nikon TiE, not a confocal microscope, therefore giving a greater depth of field.

- Please discuss the importance of the LD size differences observed in Fig 1D.

The LD sizes in Fig 1D have now been addressed, and extra text included at line 103 and 105.

- In this figure the authors mention that there are more basal level of LDs in astrocytes. In that case, please justify the use of oleic acid to stimulate more LD accumulation?

We assume the reviewer is referring to figure 2, where we show that astrocytes have a higher basal level of LDs. The use of oleic acid in figures 4 and 5 is to specifically determine the role of increasing LDs in this cell type above their basal level.

Figure 2: Treatment with dsRNA and dsDNA accumulates LDs

- The authors mention that nucleic acid stimulation drives both the generation of new LDs as well as growth of existing LDs. It is also possible that the existing LDs are breaking down during this process. Based on just differences in size, it is difficult to interpret the data. The authors could potentially address this by performing a fatty acid pulse-chase experiment as performed by Rambold et al., *Developmental Cell*, 2015.

We agree that this is an interesting question. We have now included new experiments to this manuscript using the DGAT1 inhibitor (Supplementary figure 3A), which clearly show that when the synthesis of triglycerides (the main lipid in LDs) is inhibited, the formation of nascent small LDs as well as the growth of larger LDs is absent, indicating that in figure 2, we are likely seeing the generation of new LDs. The text and methods have been adjusted accordingly.

- Fig 2D: Is this figure for dsRNA or dsDNA-treated cells?

We apologize for this omission, and this is now clearly labelled in the figure and the figure legend as dsRNA stimulation.

Figure 3: LD accumulation is transient

1. Although the authors mention that dsRNA and dsDNA-mediated LD induction coincide with the IFN levels, at 24h dsRNA-treated cell have highest IFN levels whereas LD numbers are dropping down at this time point. Please write a more accurate interpretation.

We have now toned down this statement in the text to say, 'followed a similar pattern'.

Figure 4: Increased LD numbers enhance Type I and III IFN response to viral infection

1. Increase in IFN after dsRNA and dsDNA treatment could simply be a host response irrespective of LDs. As mentioned above, performing these studies in cells lacking LDs would answer this question.

We agree with the reviewer and have now performed experiments in Figure 7F and 8F and where LD Induction is blocked via EGFR inhibition following cellular treatment with dsRNA, dsDNA or viral infection. These experiments further demonstrate that LDs are required for an efficient production of type I IFN.

2. What is the distribution of pattern recognition receptors on astrocytes and would that play a role in IFN production in response to viral antigen-stimulation?

This is a very interesting question, and recent work has demonstrated that astrocytes express a wide range of PRRs (as reviewed in (Farina et al. 2007)) and have been found to be key players in the production of interferon in the CNS which protects them from viral infection (Lindqvist et al. 2016).

3. In Fig 4D and E, if the IFN β mRNA level is not significantly different in dsDNA-treated cells, how do the authors explain the increase in IFN β protein levels after treatment with OA and dsDNA in Fig 4G compared to control?

It is well recognised that the measurement of a genes' transcriptional response is not always linked to protein output ((Koussounadis et al. 2015)); and perhaps the increase in protein output may be a result of increased translation, or enhanced mRNA lifespan.

4. In Fig 3, IFN β and lambda mRNA fold increase rises from 8h to 24 hours with 24hours being at its maximum. However, in Fig 4, especially for the dsRNA-treated cells, the mRNA fold increase is much lower at 24 h compared to 8h in the control. Why?

The fold changes of a given gene at a given time point using Q-RT-PCR, especially when looking at genes that are induced from a very low basal mRNA level, are highly influenced by cell cycle regularity, and indeed on a day-to-day basis. However, throughout this manuscript and from figure to figure we continuously show the same trend in change.

Figure 5:

1. The authors show IFN protein levels for ZIKV but not HSV? Why?

HSV-1 is known to shut-down the IFN system very rapidly following viral infection (reviewed in (Toqnarelli et al. 2019)), which in part can be seen by the very low levels of type I and III IFN mRNA in figure 5C. As such protein analysis was not performed in this instance. However, these experiments were all performed using viral mimics initially (Fig 4), for this very purpose, to ascertain the role of LDs in the absence of viral antagonism of innate immunity. Indeed, new experiments also performed in relation to an above question, demonstrate that the production of IFN protein following HSV-1 infection is below detectable levels in our hands (Figure 8F)

2. Why does ZIKV, an RNA virus, does not induce IFN protein production during infection of control cells but does in OA-treated cells? This is especially important to know since ZIKV is shown to induce LDs in Fig 1. This further weakens the argument that LDs induce IFN-mediated antiviral response.

We have now adjusted the figure legends to be able to better see that ZIKV does indeed induce IFN protein both with and without oleic acid treatment; albeit at a very low level.

3. Showing LDs numbers in ZIKV and HSV-infected cells after oleic acid treatment will help better interpretation of this figure.

We are sorry for the omission of this data, a graphical set of microscopy figures is now given as Figure 5A, to demonstrate that viral infection on top of oleic acid treatment of cells further enhances the LD numbers. The text has been updated accordingly.

4. Line 201-202 – The authors mention 'astrocytes with upregulated LDs'. However, Fig 5 does not give proof of that. LDs should be counted before concluding their role in IFN production.

As mentioned above, this is now included in Figure 5A and supplementary figure 4B.

Figure 6: Initial LD accumulation is IFN independent

1. To strengthen the IFN-independent initial induction of LDs, could the authors use IFN knockout astrocytes (from IRF7 KO mice) or any other feasible resources?

We thank the reviewer for this comment and have now included a study using IFNAR blocking antibodies on astrocytes to demonstrate that the initial first wave of LDs is indeed IFN independent (Figure 6F and Supplementary figures 5B/C). Additionally, this data is also supportive of a second IFN-dependent wave of LDs in dsRNA stimulated cells.

We have added text on line 247 that describes this:

“To fully elucidate the role IFNs were playing in LD accumulation, we utilised an antibody that could block IFN inducible LDs in the astrocytes (Supplementary Fig. 5B). To validate the bi-phasic and IFN independent LD accumulation following dsRNA stimulation, we blocked IFNAR1 in astrocyte cells and stimulated them with dsRNA and dsDNA for up to 72 hrs (Fig. 6F and Supplementary Fig. 5C). There was no difference in LD accumulation at earlier time points in cells that were treated with antibody controls compared with IFNAR1 blocking antibodies, however, when looking at the 48-hr time point following stimulation, the IFNAR antibody partially inhibited the second wave of LD induction seen in the dsRNA stimulated cells, with a small but significant drop in the number of LDs also in the dsDNA stimulation condition (Fig. 6F and Supplementary Fig. 5C). These results further support initial IFN independent induction of LDs following both dsRNA and dsDNA stimulation, with a second upregulation of LDs being dependent on IFN signalling.”

As well as on line 415 which discusses this:

“Blocking of IFNAR1 was able to significantly inhibit the second wave of LD upregulation but not completely abolish it, perhaps indicating that type-III IFNs may also play a role in this second wave of induction (Fig. 6F).”

2. Based on Fig 6E, is IFN β the most important IFN since astrocyte dsRNA media-treated Vero cells show similar LD #s as that of 1000U/ml IFN β -treated cells? What is the role of IFN λ that is shown to increase in previous figures?

We thank the reviewer for this question; and as can now be seen in the new figure 6F, it appears that type I IFN alone, is not the only driver of the second wave of virally driven LDs, and indeed it is likely that type III IFN may play a role. Further discussion surrounding this has also been added in the final discussion.

3. Most experiments in this figure are only performed n=2 questioning the statistical significance of this data.

Although these experiments are performed as n=2 biological replicates, we are in fact counting in excess of 200 individual cells to enumerate the LD numbers in each instance.

Figure 7: Role of EGFR in LD induction

1. Although the authors mention DMSO has no effect, this reviewer would like to see a vehicle control for Fig S6A,B.

We are sorry for the confusion, all of these experiments are vehicle controlled, where control is listed, it has DMSO also added to the same percentage as the treatment groups. This has now been made clearer in the methods section.

2. If possible, please change Fig S6A. The image is blurry and difficult to observe differences.

We are unsure as to why this image may appear blurry, in our document, both S6A and S6B have the same resolution.

3. Line 269-270 – Could you please clarify the precise role of EGFR?

This has now been done, and reads:

“EGFR is a receptor tyrosine kinase that is commonly upregulated in cancers but is also targeted by several viruses (De Larco and Todaro 1978). As it is known that EGFR underpins LD upregulation in some cancers, and in figure 7 we have demonstrated that EGFR could also drive virally induced LDs, we next wanted to understand the relationship between viral-induced EGFR driven LD biogenesis and the regulation of antiviral cytokines.”

Figure 8: EGFR-induced LDs impact IFN production and attenuate viral infection.

Could the authors please speculate in their discussion if LDs are induced early during viral infection resulting in IFN production which can attenuate viral replication, how are the viruses surviving during in vivo infections?

We feel that LDs are induced very early following viral infection, as indicated by our initial experiments in figure 3. However, as many viruses possess the ability to usurp early innate immune responses, and attempt to limit viral infection, it is often the role of the uninfected bystander cells to upregulate ISGs following an initial infection event, and ultimately limit viral replication. However, as a true infection is a spreading infection, there would be a dynamic equilibrium between the virus and the innate response, and this is constantly being researched in this field. We feel that this has already been discussed in the current manuscript from line 422

Inconsistencies in the paper:

• **Could the authors explain the rationale behind different MOIs for ZIKV between figures? For example: Fig 1 – MOI – 5 for ZIKV and HSV, Materials – MOI - 0.1 for HSV and Zika.**

For figure 1, we wanted to ensure that every cell was infected in the culture, to ascertain whether initial viral infection could upregulate lipid droplets. However, in most future experiments, an MOI of 0.1 was utilised to ensure a more physiological viral infection, that spreads throughout the culture over time.

• All figure legends lack several important details which are mentioned in materials and methods. Having all the details in the legend, especially times post-infection/treatment will help with analyzing the figures.

We thank the reviewer for their suggestion, and this has now been added throughout the figure legends where appropriate.

Statistical analyses:

• Although the analysis seems sound, the legend does not mention what the significance is compared to, leaving it up to the reader's interpretation.

We have now either added lines on graphs between samples that are compared throughout the figures, or alternatively if this would impair the visibility of the figures, an explanation is now added into the figure legends.

Minor revisions:

Grammatical and writing errors:

1. Introduction:

• Line 52 – Italicize bacteria names

This has been done

• Line 56 – mentions Trypanosoma cruzi infection of macrophages also induces LDs but takes 6-12 days to occur following infection. The authors should correct this sentence as reference#8 have measured LD numbers at 24 hours post-infection in cells including peritoneal macrophages.

• Line 67-68 – Italicize M. tuberculosis

This has been done

• In the introduction, the authors do not mention HCV whose relation with LDs is extensively studied.

We agree with the reviewer that HCV has an extensive history with lipid droplets, particularly in the use of them to assist in their viral life cycle. However the virus itself has not been considered to induce LDs. As such we feel that the current discussion surrounding HCV at the end of the manuscript is well placed.

- **Line 122 is quite confusing. Do the authors mean after stimulation or in unstimulated cells?**

We are not entirely sure as to exactly which part of this sentence the reviewer is referring to, however, the term 'basal' levels is referring to pre-stimulation; and the statement in regards to THP-1 macrophage cells, mentions in the text sizes do not change pre or post stimulation.

- **Line 124-125 mentions THP-1 average basal LD size did not increase with stimulation. But Line 126-137 – dsRNA stimulation 2-fold increase in THP-1 monocytes? Please clarify.**

This is correct, THP-1 macrophage LDs did not increase in size, but THP-1 monocytes did.

- **Line 173-174 – sentence is unclear. Are the authors talking about cells treated with dsDNA?**

We have altered this sentence to make this more clear

- **Line 512 - mentions 'INF' instead of 'IFN' used throughout the manuscript. Please change for consistency.**

This has been changed

Reviewer #2 (Remarks to the Author):

In this study Monson et al. describe the induction of lipid droplets (LDs) and Interferon response in relation to viral infection and its mimetics. The topic is highly interesting and the study provides informative data regarding lipid body induction kinetics. However, more mechanistic studies on LD induction in infection models are lacking, conclusions are drawn based on limited data sets and key experiments are missing. The experiments investigating the role of EGFR on LD induction are novel and appealing but lack mechanistic insights. IFN induction and LD induction upon viral infection is not novel, and the current data are insufficient to show causal relationships.

Major points

- The authors state that different types of LDs are present within their models. An additional experiment showing that those LDs are actually different, based on LD marker expression or content would make this statement more convincing.

We refer to lipid droplets in our manuscript as 'heterogeneous' based mostly on size differential, however, there have been a small handful of studies which have managed to either fractionate LDs from within a population of cells, and assess their proteome, or alternatively, use fluorescence microscopy to demonstrate that some LD resident proteins do not reside on the entire population of

LDs in a single cell (Zhang et al. 2016; Thiam and Beller 2017). However, we feel that this knowledge is outside of the scope of this manuscript.

- In figure 1abc, a co-localization of the virus with LDs is seen. The effects described in the manuscript suggest a signaling role of LDs, especially on IFN response. However, if co-localization of LDs is seen with the virus, could this mean a direct virus-LD interaction?

We thank the reviewer for these comments, however, without the use of more specific antiviral antibodies this is not possible to determine. These infections are performed with a very high MOI and making it difficult to ascertain if viral proteins are truly interacting with the lipid droplets. However, it is known that DENV positions one of its structural proteins on the LD, as do a number of other flaviviruses.

- In line 145-149 the authors describe a bi-phasal induction of LDs upon dsRNA stimulation. This seems to be based on the drop of LD numbers in one time measurement (24h, figure 3b). A further time course including multiple time points is necessary to show this is an actual bi-phasal response. Moreover, in the discussion, line 381-383, the authors conclude that this bi-phasal response was mediated by EGFR and later by IFN. However, the experiments performed are not sufficient to approve or reject this hypothesis.

We have now performed additional experiments (Figures 6F, and Supplementary figures 5B and C), using an INFR1 blocking antibody, which impedes upregulation of LDs via IFN. This new data clearly shows a biphasic response to LD upregulation following dsRNA stimulation; with the first wave being IFN independent and the latter being somewhat IFN dependent. This data has now been discussed in the final section of the manuscript as well.

- In line 181- 183 the authors conclude that presence of LDs was able to enhance IFN production. However, experiments to show a causal relationship between LDs and IFN production are lacking. Therefore this conclusion cannot be drawn based on the experiments performed.

Please see the answer to reviewer 1 above - these experiments have now been performed and are in figure 8F demonstrating when LDs are unable to accumulate (EGFR inhibited) IFN production is reduced.

- Regarding figure 5, in line 200-202 the authors state that reduction in viral load for ZIKV coincided with enhanced levels of IFN and with upregulated LDs. However, no LD quantification is present for figure 5 neither for ZIKV or HSV-1 infection in this setting, making it impossible to prove that indeed LD numbers coincide with IFN production and reduced viral load in this setting. The LD quantification of figure 1 is performed with a higher MOI, making direct comparison impossible. This also counts for figure 8, where LD quantification is missing.

We are sorry for the omission of this data set, as mentioned in response to reviewer 1's question, this data is now included in Figure 5A, and the corresponding supplementary figure 4B

- In figure 5c and 5d, the virus is still able to replicate, even in the presence of IFN and LDs, and only seems to be inhibited by addition of Oleic Acid. In this experiment it is not proven that the presence of LDs themselves have an actual inhibiting effect on viral replication.

In figure 5 the cells are infected with a virus, which in turn enhances IFN production in the cells. The purpose of the addition of oleic acid is to enhance the production of LDs in these cells, which can be seen in the accompanying figure 5A and supplementary 4B. Additionally, we have shown in figure 8, that when we inhibit the virally driven production of LD, we lower the IFN level, and enhance viral replication. We believe that these observations taken together very strongly suggest that the LD is a pivotal organelle in the production of IFN.

- In figure 5e, why is infection with the ZIKV not able to induce IFN production without Oleic Acid treatment? The authors have shown that mRNA and protein levels of IFN can differ, so the protein levels of IFN production upon HSV-1 infection should also be shown.

As mentioned above in response to the question from reviewer 1:

“We have now adjusted the figure legends to be able to better see that ZIKV does indeed induce IFN protein both with and without oleic acid treatment; albeit at a very low level.”

“HSV-1 is known to shut-down the IFN system very rapidly following viral infection (reviewed in Tognarelli et al. 2019), which in part can be seen by the very low levels of type I and III IFN mRNA in figure 5C. As such protein analysis was not performed in this instance. However, these experiments were all performed using viral mimics initially (Fig. 4), for this very purpose, to ascertain the role of LDs in the absence of viral antagonism of innate immunity.”

- In figure 6, in our opinion it is possible that the increase in LDs seen in Vero cells upon treatment with dsRNA astrocyte media could be induced by the presence of IFN in the media, as Vero still have an IFN α / β receptor. Therefore, the control treatment of Vero cells with IFN is missing.

We apologise for the omission of this data. We agree that this data should have been included and therefore this can now be found as Figure 6B.

- The conclusion in line 267-270 is not clear. Apparently, there are two mechanisms involved in the LD induction seen, one IFN mediated, and one EGFR mediated. This should be written more clearly.

We thank the reviewer for their comment and agree that this could be more clear. We have therefore added an additional and final figure to the manuscript that outlines the mechanisms described in the manuscript. This figure is included as Figure 9 and we have referenced this figure in the discussion on line 409 and 420. Additionally, the figure can be found below along with its figure legend.

Figure 9. Lipid droplets are induced to enhance an effective interferon response

Here we demonstrate that the host upregulates LDs following pathogen detection in two waves. The first wave is EGFR dependent which is independent of interferon production, and the second wave is induced via interferon production. During dsDNA stimulation, there is no second wave, and this is hypothesised to be due to a negative regulator of LD accumulation produced by cells during this response.

- Data about the importance of the described processes in vivo is lacking and is overstated in the summary.

The ability to assess the importance of virally driven LD upregulation in the outcome of viral infection in vivo is difficult, however we are currently examining a number of KO murine animal models for their ability to limit LD upregulation following viral infection without other significant impacts on the animal's physiology; given the importance of lipid metabolism in viability. However, we have now included a second in vivo infection model in figure 1F (and Supp1B) showing that the phenomenon of LD upregulation very early following viral infection is not just limited to influenza or the lung. We appreciate your concerns and have now toned down our statements in the summary section of this manuscript.

Minor points

- Bodipy 409/505 is used in the study. Please state the company, since the commonly used dye to stain LDs is Bodipy 493/503.

We are very sorry for this oversight, and the correct bodipy has now been listed in the methods and figure legends.

- In the summary, line 33-35, the authors state that LD induction is IFN independent. However, in the discussion they explain that the second part of the bi-phasal induction might be IFN dependent. Please be consistent.

We have now altered the summary text to say, 'Initial virally driven LD induction....'

- Please add bars when showing significance to allow the reader to follow which results were directly compared and significant.

Please see the repeated answer to reviewer 1's question below:

"We have now either added lines on graphs between samples that are compared throughout the figures, or alternatively if this would impair the visibility of the figures, an explanation is now added into the figure legends."

- Please elaborate why these specific infection models (virus and cell type) were chosen for the current study.

Please see the below repeated answer from a reviewer 1 question:

Multiple cell types are utilised in the first 2 figures of this manuscript, however, to fully elucidate the mechanisms and dynamics of LD upregulation following viral infection, we chose to continue the work in the manuscript using astrocytes. The following text has now been added to the manuscript at line 153 "Astrocyte cells were chosen to further examine the role of LDs in viral infection due to their extensive upregulation of LDs following viral stimulation and infection. Additionally, astrocytes are known to be rapid producers of an effective antiviral response (Lindqvist et al. 2016)."

The choice of viruses was based on ensuring that we covered both an RNA (Influenza and ZIKV) virus, as well as a DNA virus (HSV-1). However, we thank the reviewer for the suggestion of including a flavivirus that is known to utilise lipid in its replication cycle, and have now included both in vivo (Figure 1E and Sup-1B) and in vitro (Fig 1B,C and Sup-1A) analysis of upregulated LD regulation following DENV viral infection in mice and astrocytic cell lines.

- Additional information about cell viability would be interesting to add, since LDs can also be seen in stressed cells.

We agree that the intersection between cell viability and LD regulation is an interesting area, however, is outside of the scope of this manuscript. In general astrocytes throughout this study and others were seen to and are known to be generally refractory to death by viral infection. No significant cell death

was observed during these studies; however, an examination of the role of virally driven LD induction and cell death pathways may be interesting to explore in the future in an alternate cell type.

- Please add the labels of the antibody and color shown in the confocal pictures in the figure itself.

This has now been added

- In figure 1a the more virulent PR8 and less virulent X-31 strain of influenza are used. Could you elaborate why the X-31 seems to infect the cells better in this model (more cytoplasmic staining)?

The virulence of these strains refers to the disease outcome in animals, rather than the infection rate in in vitro studies; however, we do not believe that the infection level is higher in the X-31 infected cells.

- In figure 1b the ZIKV is stained using two antibodies, one of which for dsRNA. Could you please explain why, and also which staining is visible in which color and which picture since only three colors are seen in figure 1b instead of four.

We are sorry for the confusion; both antibodies stain dsRNA and are generally used in combination with each other to allow good detection of all dsRNA in the cell. This is stated in the methods, and hopefully is now clearer in the figure legend.

- In line 94-95 it is mentioned that astrocytes have a high basal level of LDs (approximately 15 per cell). In the confocal pictures this is not clearly visible. Please add a clear confocal image showing the basal level more clearly.

We apologise for the quality of the figure; this has now been adjusted. Additionally, a similarly high basal level of LDs in astrocytes can also be seen in figure 2A.

- In figure 1d a LD staining in lung tissue is shown, are those really LDs, since the staining seems not be inside cells? Please state what is stained in blue.

The figure legend now states that the blue staining is nucleic acid (DAPI stained). In regards to the Large LDs, macrophages are known to be recruited to the bronchioles during influenza infection, and these cell types are known to form very large LDs. Additionally, Bodipy only stains dense pockets of neutral lipid, such as those found in cellular lipid droplets.

- Please describe if the LD count in virus infected samples is performed for all cells present in the sample or only for the infected cells within that sample.

If the reviewer is referring to Figure one, then the LDs were counted across all cell, however an MOI of 5 was utilised to ensure 100% cellular infection.

- In line 22, the conclusion is not correct, since only for THP-1 cells the size of the LDs does not change, for the other cells it does.

We are unsure as to the issue with these particular sentences (taken from line 122 - we presume the reviewer is referring to this line as line 22 has no text). The text below is consistent with what the review would like us to mention. We apologise if we have misunderstood the comment.

“The average basal size of LDs was consistent across most cell types, with a diameter range of 280-400 nm (Fig. 2C); however, THP-1 macrophages had a starting average basal LD size of 3100 nm, which did not increase following stimulation with either dsRNA or dsDNA. In contrast, all other cell types had an increased average LD size at 8 hours following dsRNA stimulation, ranging from a 2-fold increase in THP-1 monocytes to a 5.3-fold increase in HeLa cells, with similar size increases observed following dsDNA stimulation also (Fig. 2C).”

- In Supplementary figure 2, please describe what is meant with “serum replacement medium”. Also the sample names of the confocal pictures in Supplementary 6b is unclear and does not seem to match with the quantification. Is “replenished serum” with 2 or 10% FCS?

Serum replacement medium is outlined in the methods and is an artificial medium that does not contain growth factors, steroid hormones, glucocorticoids, cell adhesion factors, detectable Ig and mitogens. We have now added the company name in the figure legend to more definitively outline that this is a commercial product.

In response to the second query, we have now adjusted the quantification names on the graph to more closely match the microscopy pictures.

- In line 181- 183 the authors state a high amount of LD leads to IFN production. However, in figure 4, the combination of Oleic Acid with dsDNA does not lead to an additional increase in LD numbers compared to dsDNA alone, but it does lead to an increase in IFN production on protein level. How can this be explained?

In Figure 4C, we do see a small but significant rise in LD number per cell between oleic acid treated astrocytes alone, and oleic acid treated + dsDNA. However, we believe that our manuscript data clearly implicates the role of the lipid droplet in driving an effective IFN response in the cell, however it is unlikely to be the only mechanism at play that drives an increase in IFN production following detection of viral nucleic acid by Pattern recognition receptors.

- As the authors state in line 200-202 that reduction in viral load and enhanced IFN production coincide with LD number, it would be interesting to see the viral load of figure 5c and 5d at 72h as it is the time point that LD numbers are returning to basal levels again in the dsRNA/dsDNA setting.

Although we agree that this is an interesting question, the ability of the two viruses to augment the interferon signaling pathways, and in the case of ZIKV deplete LDs at later time points (Supp S9), the dynamic between LDs and viral replication would be difficult to dissect.

- The graph of figure 7b has a mistake in the labeling of the last condition.

This has now been corrected.

Reviewer #3 (Remarks to the Author):

In the manuscript by Monson et al. the authors study the impact of lipid droplet (LD) formation upon viral infection and show evidence that type I interferon (IFN) production relies on this process. They show data with a DNA virus (HSV-1 and an RNA virus (ZIKV), which highlights the importance of this pathway during viral infections.

The proposed mechanism is that as viruses infect cells, LDs are stimulated and IFN is produced by infected cells. The authors show that there is an increase in LDs and type I IFN in the presence of oleic acid in the context of infection. On the other hand, they show that pretreatment with an EGFR inhibitor (which blocks LD biogenesis) results on an increase in virus production and a decrease in LD production and IFN production. The authors have a previous publication showing the effects of oleic acid on LD formation and Sendai virus infection. In the present manuscript the authors expanded those studies to test also a DNA virus and use more relevant primary systems, such as astrocytes, although it is not fully elucidated.

Major comments:

1. Overall, the manuscript is well written, even though at times it is difficult to understand the rationale for the proposed experiments until later on. More details and background information at the beginning of some of the sections would be helpful (see below).

We thank reviewer 3 for their suggestion, and where possible we have tried to include extra information to introduce proposed experiments, see below for more specific answers.

2. From Figure 3 onwards, the authors switched back and forth between infectious virus to DNA/RNA and then back to infectious virus. It is not well explained why do they do such change. The following questions arise:

The use of viral mimics allows us to dissect the underlying mechanisms of virally induced LDs without viral antagonism, given that most viruses are able to inhibit pathways of interferon production in cells once established. As such, we have now added the following text at lines 116 to orientate the readers as to why we have utilised both virus and viral mimics.

“Additionally, the use of these viral mimics may allow the further dissection of the mechanisms of LD induction following activation of viral PRRs in the absence of viral antagonism.”

Could direct DNA/RNA stimulation lead to different times of LD production compared to a real infection?

It is plausible that this could be the case, although we know from the work in this manuscript that induction from both sources leads to an enhancement of interferon, so we feel that despite the stimulus, the outcome of the LD upregulation is similar. Having said this, it is well known that LDs are heterogeneous in nature, and a true comparison would need to include both proteomic and lipidomic analysis of cellular LDs.

Is it possible that the outcome of LD production would be different in the context of cells dying due to infection?

It is possible that this may be the case, however this work seeks to understand the initial host cell response to viral infection, and the consequence of that response. Cell death is more likely to occur at a much later time point, typically days following initial viral infection, depending on the virus. Additionally, not all viruses are lytic. We believe that the interplay between the host cell and the virus is likely to be very complex in nature in regards to LD production, especially in light of the fact that some viruses are able to usurp host cell lipid production. These questions are complex, and we hope to unravel them in time, however this study seeks to introduce the concept that the host cell induces LDs following viral infection as a means of amplifying the host cell interferon production.

Is the amount of DNA/RNA they use to stimulate comparable to the amounts that would be produced in a cell upon infection? Is it possible you are overstimulating?

The amount of dsDNA and dsRNA we have used in these experiments is in line with what is utilised in the literature. Additionally, if you refer to figure 2A you will see that the intracellular stained dsRNA is very limited in comparison to the dsRNA present in virally infected cells as seen in figure 1; so in fact we are likely to be getting less dsRNA in our cells initially in comparison to a true viral infection.

3. Fig. 6E: It would be useful to quantify the amount of IFN they had in the conditioned media previous to addition to naive astrocytes. Would also help for understanding what is going on in the DNA stimulated conditioned media. In that sense, maybe spiking the DNA stimulated conditioned media with IFN and having a “positive control” media with known amount of IFN could help too.

Unfortunately, we did not do this prior to the experiments being run, however as can be seen in figure 4F and 4G, we do see IFN present in the media from both dsRNA and dsDNA stimulated astrocytes in repeated experiments. We agree with the reviewer that the lack of a second wave of IFN produced LDs in dsDNA stimulated cells is an interesting phenomenon that requires further investigation; however, we do not feel that it impacts the main message of this manuscript.

4. Fig. 7B: Why are they showing LD size instead of LD count? Can't really compare with following panels (7D, for instance). From the microscopy it looks that while there is LD production, it is not much as compared to observed in Fig 2A. Is it possible that indeed the AACOCF₃ is inhibiting LD production as well in the presence of virus too? Why would the enzyme only inhibit PLA in the

context of mock infection but not in the context of viral infection (assuming there is only way LD biogenesis pathway)? The control with oleic acid here would be a good addition, like in Figure 7C.

We apologize about the confusion over this figure as it was mislabelled as LD size and not LD number. We have now changed this to reflect the data which is showing LD numbers. Figure 7 explores the different mechanisms of LD upregulation. There are indeed several ways in which LDs can be induced, however this is a very novel research area and we are hoping to shed some light on this from our study. We are demonstrating in this figure that PLA₂ does not control LD biogenesis to virally driven LDs and that this is actually controlled by the EGFR/PI3K pathway. As requested by Reviewer 1, we have also included an additional figure in the supplementary figures (Supplementary figure 3A), which clearly demonstrates that when the synthesis of triglycerides (the main lipid in LDs) is inhibited via the DGAT enzymes, the formation of nascent small LDs as well as the growth of larger LDs is absent, during stimulation. The text and methods have been adjusted accordingly.

5. The in vivo studies are quite limited.

We have now expanded our in vivo analysis of lipid droplet upregulation in the murine brain following dengue viral infection. This is now included in Figure 1E (and supplementary figure 1B), and the matching in vitro analysis of dengue viral driven LD's in astrocytes is now included in figure 1B and 1C, as well as supplementary figure 1A.

Minor comments:

1. On multiple occasions they use acronyms that were not properly introduced. Line 34, 77, 89, 242, 485

These abbreviations have now been properly introduced.

2. They have to homogenize criteria for presenting references, either "9,10" (line 60) or "5-7" (line 55).

We have followed the guidelines for Nature Communications on referencing, with 2 references listed with a comma and 3 or more with a dash.

3. Figure 4G: Could they specify exactly what are we seeing in panels F and G? They look exactly the same in the fig. Ideally, one should be able to follow by reading the manuscript and looking at the figures what are the authors trying to show. As it is, it's not possible to do so without going to the figure legends.

The difference between Figure 4 G and 4F is the stimulants (dsRNA and dsDNA). These are labelled on the figures so we are not sure how we can make this any clearer to the reader.

4. Figure 4 G and F and Fig. 5: Overall, it is not clearly explained/understood in the manuscript why do they do some of the experiments until later on. For instance, not very clear why do they treat with oleic acid? To over-stimulate production of LDs with what purpose? We know already that LDs are over produced in the presence of virus or dsDNA/dsRNA so why the need to add oleic acid? And we know there is IFN stimulation form presence of dsRNA/dsDNA or from LDs from literature or their own previous work. Then, when you reach in the text Figure 5, it becomes more evident they are trying to assess the antiviral effect of LDs.

We thank the reviewer for their comment; however, we feel that we have introduced this concept at the beginning of figure 4 (where we first use oleic acid to drive LD accumulation). See below for text already at the start of this section on line 173:

“We have previously demonstrated that loss of cellular LDs impacts the host cell response to viral infection in vitro¹⁹. To determine if the upregulation of LDs following viral infection plays an antiviral role in the cell, we initially established a LD induction model in the primary immortalised astrocytes. Addition of oleic acid to cells has previously been shown to enhance LDs minutes following treatment in Huh-7 cells²⁰.”

We use oleic acid to drive the production of LDs before viral infection/stimulation to access the importance of the LDs in enhancing the interferon production. We show in figure 4F and 4G that simply treating cells with oleic acid and hence upregulating LDs does not drive IFN responses on its own (with a small response seen at the mRNA level in Fig 4D and 4E) and we only see a significant difference in IFN output following stimulation or infection.

5. Line 267 – Why did you chose the 16-hour time point? It would have been more informative to maintain the same ones.

The 16-hour time point was chosen as a potential maximal protein level for IFN, based on the mRNA results, and knowing the short half-life of this cytokine.

6. Fig 7 - They are missing cell viability assays to show drugs at the concentration used are not killing the cells.

We thank reviewer 3 for their concern over the viability of our cells following various treatments. We have performed these experiments at a range of different concentrations with us picking the lowest possible concentrations for these experiments and have always included controls for the treatments we have used (cells just receiving the treatments as well as vehicle controls). All of the treatments that we have utilised throughout this manuscript are well described and heavily established for cell culture use.

7. Line 521 – This should say size instead of number.

This has now been corrected.

Reviewer comments, second round -

Reviewer #2 (Remarks to the Author):

The revised version of the manuscript has significantly improved by the addition of an in vivo model and mechanistic experiments. Please have a look at the legend of fig 8F, which timepoint is shown?

Reviewer #3 (Remarks to the Author):

NCOMMS-20-02877

Title: Intracellular Lipid Droplet Accumulation Occurs Early Following Viral Infection and Is Required for an Efficient Interferon Response

Summary: In the manuscript by Monson et al. the authors study the impact of lipid droplet (LD) formation upon viral infection and show evidence that type I interferon (IFN) production relies on this process. They show data with a DNA virus (HSV-1 and an RNA virus (ZIKV), which highlights the importance of this pathway during viral infections.

The proposed mechanism is that as viruses infect cells, LDs are stimulated and IFN is produced by infected cells. The authors show that there is an increase in LDs and type I IFN in the presence of oleic acid in the context of infection. On the other hand, they show that pretreatment with an EGFR inhibitor (which blocks LD biogenesis) results on an increase in virus production and a decrease in LD production and IFN production. The authors have a previous publication showing the effects of oleic acid on LD formation and Sendai virus infection. In the present manuscript the authors expanded those studies to test also a DNA virus and use more relevant primary systems, such as astrocytes, although it is not fully elucidated.

Decision:

The authors response to our comments and the state of the revised manuscript satisfy our major concerns. However, there were a few additional minor concerns that should be addressed prior to publication.

Minor Concerns:

- Please provide some explanation of why doesn't the size of LDs in THP-1 monocytes change following dsRNA or dsDNA stimulation (Figure 2C)
- Lines 550-552: "Primary immortalised astrocyte cells were treated with 500µM oleic acid for 16 hours prior to stimulation with dsDNA or dsRNA and analysed for LD numbers". Please provide information about how long after stimulation with viral mimics were these cells analyzed for LD numbers in the representative figure (figure 4C)
- The scale on the y-axis for figures 4D and 4E make the data difficult to interpret. Please modify.
- Lines 359-360: "LDs were induced upon infection with ZIKV, influenza or herpes simplex virus-1" zika virus is abbreviated but not influenza or herpes simplex-1 virus. Please correct for consistency.
- Please indicate if LD formation is transient in response to OA treatment as it is shown in response to viral mimics. Also, please indicate somewhere in the text if 16 hours is the peak of LD number formation in response to OA (figure 4A)
- Following LD number analyses in response to viral mimics (Figure 3A and 3B) and OA (Figure 4A and 4B) it would make more sense to use the same kinetics for further experiments (figure 4C) that assessed co-stimulatory effects of OA and viral mimics on LD biogenesis as well. One explanation for the lower levels of IFN mRNA levels (Figure 4D and 4E) could be the fact that these cells were analyzed 24- or 40-hours post OA treatment.

REVIEWER COMMENTS

Reviewer #2 (Remarks to the Author):

The revised version of the manuscript has significantly improved by the addition of an in vivo model and mechanistic experiments. Please have a look at the legend of fig 8F, which timepoint is shown?

We apologise that these details were overlooked in the figure legend. We have now added the below text:

“(f) IFN protein levels from these experiments were analysed via ELISA for IFN- β and IFN- λ protein at 16 hrs post infection.”

Reviewer #3 (Remarks to the Author):

NCOMMS-20-02877

Title: Intracellular Lipid Droplet Accumulation Occurs Early Following Viral Infection and Is Required for an Efficient Interferon Response

Summary: In the manuscript by Monson et al. the authors study the impact of lipid droplet (LD) formation upon viral infection and show evidence that type I interferon (IFN) production relies on this process. They show data with a DNA virus (HSV-1) and an RNA virus (ZIKV), which highlights the importance of this pathway during viral infections. The proposed mechanism is that as viruses infect cells, LDs are stimulated and IFN is produced by infected cells. The authors show that there is an increase in LDs and type I IFN in the presence of oleic acid in the context of infection. On the other hand, they show that pretreatment with an EGFR inhibitor (which blocks LD biogenesis) results on an increase in virus production and a decrease in LD production and IFN production. The authors have a previous publication showing the effects of oleic acid on LD formation and Sendai virus infection. In the present manuscript the authors expanded those studies to test also a DNA virus and use more relevant primary systems, such as astrocytes, although it is not fully elucidated.

Decision: The authors response to our comments and the state of the revised manuscript satisfy our major concerns. However, there were a few additional minor concerns that should be addressed prior to publication.

Minor Concerns:

• **Please provide some explanation of why doesn't the size of LDs in THP-1 monocytes change following dsRNA or dsDNA stimulation (Figure 2C)**

In Figure 2 the size of LDs does actually increases following dsRNA and dsDNA stimulation of THP-1 monocytes, however, the LDs within THP-1 macrophages do not. We have now added to our explanation in the discussion (lines 381-387), and the text reads:

Furthermore, the average size of LDs in different cell types was also shown to increase with the exception of LDs from THP-1 macrophages (a cell type that already displays a large average size of LDs without prior stimulation), perhaps demonstrating that there is an optimal size range for LDs in respect to their functional importance following a viral infection. Alternatively, as the LD sizes calculated are an average, it is still plausible that resident LDs in THP-1 macrophages have grown in size following stimulation, however a larger influx in nascent LDs may have reduced the overall average size for this particular cell type.

*NB** on re-examination of Fig 2, we noticed that the text boxes in Fig 2B which display the statistical significance of the graph bars were misaligned, making it appear as a *** in 2 places, rather than a ****. We have now corrected this.*

- **Lines 550-552: “Primary immortalised astrocyte cells were treated with 500µM oleic acid for 16 hours prior to stimulation with dsDNA or dsRNA and analysed for LD numbers”. Please provide information about how long after stimulation with viral mimics were these cells analyzed for LD numbers in the representative figure (figure 4C)**

We apologise that this information was not included. We have now changed the wording of the sentence to include this as below (line 545):

“Primary immortalised astrocyte cells were treated with 500µM oleic acid for 16 hours prior to stimulation with dsDNA or dsRNA for 8 hrs LD numbers were analysed.”

- **The scale on the y-axis for figures 4D and 4E make the data difficult to interpret. Please modify.**

We have now altered the scale bars on the y axis of figures 4D and 4E to have similar maximum values, hoping that this will make comparisons between the two time points easier.

- **Lines 359-360: “LDs were induced upon infection with ZIKV, influenza or herpes simplex virus-1” zika virus is abbreviated but not influenza or herpes simplex-1 virus. Please correct for consistency.**

This has now been changed in the text so that all viruses are abbreviated. (Line 354)

- **Please indicate if LD formation is transient in response to OA treatment as it is shown in response to viral mimics. Also, please indicate somewhere in the text if 16 hours is the peak of LD number formation in response to OA (figure 4A)**

LD upregulation from oleic acid treatment of cells does not follow the same peak as we have demonstrated following stimulation/infection in cells, however, this upregulation is still transient in nature, starting to decline around 3-4 days post OA treatment (Fujimoto et al 2007; Journal of lipid research). We have chosen a 16 hr time point in this case as LDs are significantly upregulated from control treated cells, and our purpose is to investigate the differences between OA induced versus virally induced LDs. The text now reads:

Addition of oleic acid to cells has previously been shown to enhance LDs minutes following treatment in Huh-7 cells, with LDs remaining upregulated for 3-4 days following addition of oleic acid to the media.

- **Following LD number analyses in response to viral mimics (Figure 3A and 3B) and OA (Figure 4A and 4B) it would make more sense to use the same kinetics for further experiments (figure 4C) that assessed co-stimulatory effects of OA and viral mimics on LD biogenesis as well. One explanation for the lower levels of IFN mRNA levels (Figure 4D and 4E) could be the fact that these cells were analyzed 24- or 40-hours post OA treatment.**

For figure 3A and 3B we have analysed a time course of LD upregulation following dsRNA and dsDNA stimulation where we identified 8 and 24 hours as the peak production times for type I and III IFN, depending on the stimulus. In figure 4, we wished to examine whether the LD itself, as an organelle platform, could enhance the cellular response to dsRNA and dsDNA viral mimics; and as is seen in Figure 4D and 4E, this was the case. It is for this reason, that a time point of 16 hours was chosen, following oleic acid stimulation, to ensure we were seeing a large upregulation of LDs in the cytoplasm of cells. We believe that the reviewer is asking about the lack of induction of type I and III IFNs in the control treated cells of Fig 4D and 4E, which are effectively observed at 24 and 40 hrs post oleic acid addition. However, whether or not oleic acid can drive an interferon response was not the focus of these experiments but is however an interesting point in itself; but we believe, one outside the scope of this manuscript. In fact, extensive work in our lab is currently

underway observing the proteomic and lipidomic landscape of these LDs, where we will comparatively look at oleic acid driven LDs and their cargo, versus virally driven LDs.

Reviewer comments, third round -

Reviewer #3 (Remarks to the Author):

The revised version of the manuscript by Monson et al. has incorporated all the comments raised by the previous reviews.

Congratulations to the authors for their great work and interesting results.